# Transient growth factor expression via mRNA in lipid nanoparticles promotes hepatocyte cell therapy in mice

Anna R. Smith [1], Fatima Rizvi [1], Elissa Everton [1], Anisah Adeagbo[1], Susan Wu [1], Ying Tam [2], Hiromi Muramatsu [3], Norbert Pardi [3], Drew Weissman[4] & Valerie Gouon-Evans [1] ✉

Primary human hepatocyte (PHH) transplantation is a promising alternative to liver transplantation, whereby liver function could be restored by partial repopulation of the diseased organ with healthy cells. However, currently PHH engraftment efficiency is low and benefits are not maintained long-term. Here we refine two male mouse models of human chronic and acute liver diseases to recapitulate compromised hepatocyte proliferation observed in nearly all human liver diseases by overexpression of p21 in hepatocytes. In these clinically relevant contexts, we demonstrate that transient, yet robust expression of human hepatocyte growth factor and epidermal growth factor in the liver via nucleoside-modified mRNA in lipid nanoparticles, whose safety was validated with mRNA-based COVID-19 vaccines, drastically improves PHH engraftment, reduces disease burden, and improves overall liver function. This strategy may overcome the critical barriers to clinical translation of cell therapies with primary or stem cell-derived hepatocytes for the treatment of liver diseases.

Every year, there are 13,000 to 15,000 individuals in the United States on the liver transplantation waitlist. Unfortunately, thousands of these patients each year become too sick to undergo this life-saving procedure or die while awaiting it[1]. Given the scarcity of donor organs, ~150 patients in the world with over 25 various diseases, including inborn errors of metabolism, acute liver failure, bile duct disorders, and chronic cirrhosis, have received primary human hepatocyte (PHH) transplantation as a treatment and temporary solution to bridge towards liver transplantation[2–6]. While the clinical outcomes of this procedure show that it is safe, the utilization of PHH therapy remains constrained by poor engraftment efficiency, risk of graft rejection and subsequent loss of long-term benefit, and limited availability of hepatocytes[2,7–11]. PHH transplantation has also been explored in numerous preclinical mouse models of metabolic disorders affecting hepatocytes, acute liver failure, and cholangiopathies. However, high levels of liver chimerism are achieved only when paired with extreme genetic, chemical, pharmacological, or surgical preconditioning techniques that are not clinically translatable nor relevant models of human liver diseases[12–22]. These studies created mice with humanized livers that have mainly served as models for studying drug design, viral infections, and metabolism. They also taught valuable lessons that can be leveraged to promote PHH transplantation as a viable cell therapy and alternative to liver transplantation. Based on these clinical and preclinical studies, the consensus in the field is that successful proliferation of PHH grafts and meaningful liver repopulation require regenerative stimuli and a growth advantage for the transplanted cells[10], two processes that we investigated here to better enable PHH therapy for liver diseases.

[1]Department of Medicine, Section of Gastroenterology, Center for Regenerative Medicine, Boston University Chobanian & Avedisian School of Medicine & Boston Medical Center, Boston, MA, USA. [2]Acuitas Therapeutics, Vancouver, BC, Canada. [3]Department of Microbiology, University of Pennsylvania Perelman School of Medicine, Philadelphia, PA, USA. [4]Department of Medicine, University of Pennsylvania Perelman School of Medicine, Philadelphia, PA, USA. ✉e-mail: valerige@bu.edu

Regenerative stimuli for PHH transplantation have been induced via preoperative portal vein occlusion or partial hepatectomy[5,10,21]. Not only are the regenerative stimuli from these procedures short-term, but the procedures themselves pose a risk to patients. Alternatively, here we introduce the use of nucleoside-modified mRNA encapsulated in lipid nanoparticles (mRNA-LNP) to deliver regenerative stimuli. We have previously validated this approach as efficient, safe, non-integrative, and liver-targeted when the LNP were injected intravenously, enabling transient and robust protein expression in the murine liver[23,24]. mRNA-LNP safety in humans has been validated with current cancer immunotherapy[25] and mRNA-LNP-based COVID-19 vaccines[26], as well as in many clinical trials aiming to treat infections, cancer, and genetic disorders[27]. The regenerative stimuli we deliver are two human hepatocyte mitogens—hepatocyte growth factor (HGF) and epidermal growth factor (EGF). The HGF/cMET and EGF/EGFR axes, often referred to as the pillars of liver regeneration, are essential during liver development and naturally induced following liver injury in the regeneration process[28–31]. This is an efficient and translatable way to provide controlled expression of supportive factors to PHHs post transplantation.

Hepatocyte senescence, measured via expression of cell cycle inhibitor p21, is a consistent hallmark of chronic and acute human liver diseases, including nonalcoholic steatohepatitis with and without cirrhosis, viral hepatitis with and without cirrhosis, primary sclerosing cholangitis, primary biliary cirrhosis, autoimmune hepatitis, alcoholic steatohepatitis with and without cirrhosis, acute liver failure, acetaminophen overdose, and inborn errors of metabolism like alpha-1 antitrypsin deficiency[32–35]. Based on the liver disease and its progression, the percentage of p21 positive hepatocytes can reach up to ~90%[32,33]. This phenomenon is usually considered detrimental to overall liver health but puts such patients in a unique position where their liver environment may provide an inherent advantage to healthy donor PHHs that are readily proliferative if given the right cues. Therefore, in this study, we leveraged this observation by refining two pre-existing mouse models of liver injury, an acute and a chronic model, in which we recapitulated compromised hepatocyte proliferation via a single injection of AAV8-TBG-p21 that induces expression of p21 under the hepatocyte specific thyroxine binding globulin (TBG) promoter[36] (Supplementary Fig. 1). In these more clinically accurate liver injury mouse models, our study demonstrates that stimulating key regenerative pathways in transplanted PHHs using human HGF and human EGF (from now on referred to as HGF and EGF) delivered with mRNA-LNP drastically improves PHH engraftment and restores liver function (Supplementary Fig. 1).

## Results

### Validation of the mitogenic effects of HGF + EGF mRNA-LNP on hepatocytes in the refined clinically relevant p21/NSG-PiZ chronic liver disease model

As a chronic human liver disease model, we used the NSG-PiZ mouse that recapitulates alpha-1 antitrypsin deficiency (AATD) associated liver disease. These mice are transgenic for the human mutated PiZ allele on immunodeficient NOD scid gamma (NSG) background, allowing them to tolerate xenotransplantation[21]. These mice have heterogeneous accumulation of cytoplasmic misfolded Z-AAT globules in hepatocytes and develop fibrosis with age, similar to human AATD patients[21,33]. PiZ mice have been shown to support the engraftment of healthy donor mouse hepatocytes. While PHH engraftment has been observed in NSG-PiZ mice, liver repopulation was modest in comparison to mouse-to-mouse transplantation. Significant PHH engraftment was achieved only with drastic preconditioning methods such as partial hepatectomy, monocrotaline injections, or anti-CD95/FAS antibody treatment, which do not reflect clinically relevant approaches[21]. Instead, here we refined this injury model to render it more clinically relevant by mimicking p21-associated hepatocyte arrest seen in AATD liver disease patients[33], and tested whether mitogen

mRNA-LNP treatments improved PHH therapy. To do so, we administered one dose of AAV8-TBG-p21 virus that specifically induces p21 expression in hepatocytes, presumably for the lifespan of the mouse as previously reported[36,37]. One week after AAV8 injection, nuclear p21 protein was detected in murine hepatocytes (Fig. 1a). We quantified p21+ hepatocytes to be on average 15% in the control AAV8-TBG-Null group and 87% in the AAV8-TBG-p21 group (Fig. 1b). This recapitulates what is observed in human AATD patients that have a range of p21+ hepatocytes up to ~90%, with an average of 12.3% for heterozygous and homozygous patients combined[33]. We validated the efficiency of the mRNA-LNP delivery system to express protein in the NSG-PiZ diseased livers, as previously shown in healthy livers[23], by administering one dose of eGFP mRNA-LNP in NSG-PiZ mice. eGFP was efficiently expressed 5 h after injection in almost all hepatocytes regardless of the presence of the cytoplasmic disease-associated Z-AAT globules (Fig. 1c). We confirmed that there was no hepatotoxicity related to an mRNA-LNP injection, demonstrated by no change in liver alanine aminotransferase (ALT) enzyme levels 48 h following control firefly luciferase (Luc) mRNA-LNP treatment (Supplementary Fig. 2a). We also observed no hepatocyte apoptosis via TUNEL staining (Supplementary Fig. 2a). Then, the kinetic of expression of mitogens HGF and EGF were evaluated following co-injection of HGF + EGF mRNA-LNP and analyzed 5, 24, 48, and 72 h later. HGF and EGF proteins were abundantly detected in the liver tissue of treated mice, while absent in untreated mice at 0 h (Fig. 1d, Supplementary Fig. 2b, c). Similarly, as HGF and EGF are secreted proteins, circulating serum HGF and EGF peaked at 24 h after administration and decreased by 48 and 72 h (Fig. 1e, f). This general kinetic of mRNA-LNP-induced protein expression in the mouse liver has also been characterized previously by our lab[23,24]. These data demonstrate that mRNA-LNP mediated HGF and EGF expression is robust and transient in NSG-PiZ mice. Next, we explored the functions of HGF and EGF to induce hepatocyte proliferation and p21 to halt it. NSG-PiZ mice were either administered AAV8-TBG-p21 or no virus. One week later, the mice were given either HGF + EGF mRNA-LNP or control Luc mRNA-LNP treatment, and were analyzed 48 h later, when HGF + EGF mRNA-LNP mediated hepatocyte proliferation is known to be the highest[23,24]. Mice were injected with 5-Ethynyl-2'-deoxyuridine (EdU) 2 h prior to sacrifice to mark cells that were proliferating during that short window (Fig. 1g). In the absence of p21 injection, HGF + EGF mRNA-LNP induced significant hepatocyte proliferation in comparison to control Luc mRNA-LNP, as demonstrated by significantly increased EdU+ hepatocytes on liver sections (Fig. 1h, i). In contrast, AAV8-TBG-p21 injection one week prior to mRNA-LNP treatment blocked the proliferation induced by HGF + EGF mRNA-LNP (Fig. 1h, i). Altogether, we have established a refined chronic liver injury model recapitulating AATD-associated liver disease harboring both accumulation of polymeric Z-AAT in hepatocytes and nuclear p21 expression in a subset of hepatocytes, as seen in AATD patients[33]. We demonstrated that in this more clinically accurate non-proliferative host liver context, HGF + EGF mRNA-LNP driven host hepatocyte proliferation is impaired and, thus, suggests that HGF + EGF mRNA-LNP treatment may be a therapeutic strategy to specifically promote the proliferation of transplanted healthy PHHs.

### HGF + EGF mRNA-LNP treatment promotes PHH engraftment by increasing their survival and proliferation in the p21/NSG-PiZ mouse model

To dissect the mechanism of how HGF + EGF mRNA-LNP promotes PHH engraftment and to investigate the need for p21 expression in host hepatocytes to synergize the effect of HGF and EGF, we analyzed 4 distinct groups of mice (Fig. 2a). The control group was given Luc mRNA-LNP injections only. Single condition groups were given AAV8-TBG-p21 only or HGF + EGF mRNA-LNP only. The combined condition group was given both p21 and HGF + EGF injections. One million PHHs were transplanted as a single cell suspension via intrasplenic injection

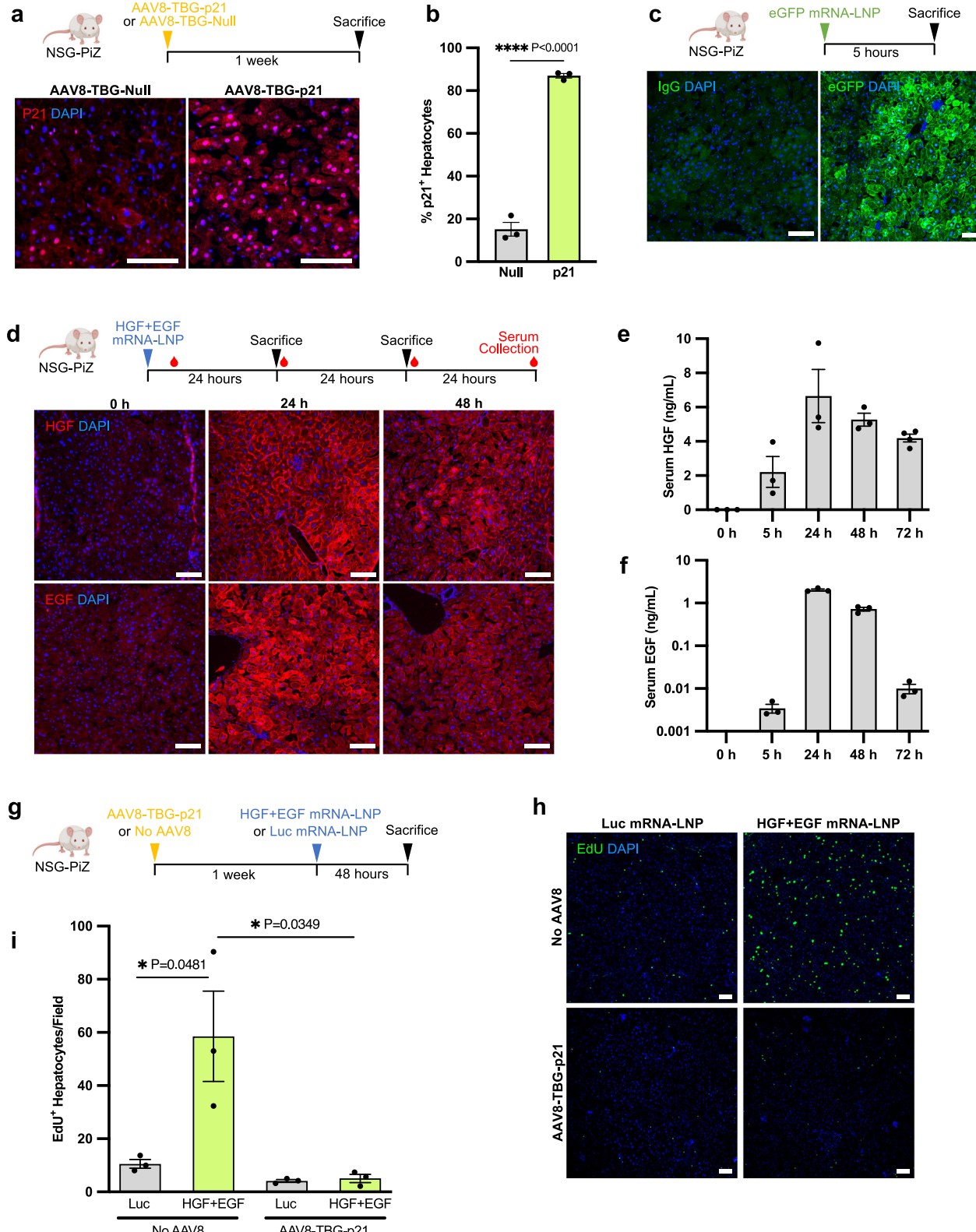

as routinely performed in mice[38]. A single dose of AAV8-TBG-p21 was injected one week prior to cell transplantation, while mRNA-LNP were administered 5 h before transplantation and twice weekly for two weeks before mice were sacrificed (Fig. 2a). We chose to analyze cell engraftment during the first 2 weeks post transplantation, early time points that are often neglected in other studies. This helps to specifically examine the benefit of HGF + EGF mRNA-LNP not only on cell

proliferation over a short time, but also on cell survival upon initial transplantation. Indeed, improving cell survival is key for efficient and successful cell therapy. Transplanted PHHs were tracked in mouse livers via co-immunostaining using a Ku80 antibody that marks human nuclei[39] and a human-specific albumin (hALB) antibody. hALB was also detected via ELISA on the serum, as functional PHHs are expected to express and secrete albumin. In the control group, mostly singlet and

**Fig. 1 | HGF + EGF mRNA-LNP induces hepatocyte proliferation in vivo, which is blocked by AAV8-TBG-p21 expression. a** Representative images of immunofluorescence staining for p21 in NSG-PiZ livers 1 week after IV administration of AAV8-TBG-p21 or AAV8-TBG-Null vectors. **b** Quantification of percent p21⁺ hepatocytes, identified by morphology. *n* = 3 mice per group. At least 3 images were averaged per mouse, each image from a different liver lobe. Gray−null, green−p21 group. **c** Representative images of immunofluorescence staining for eGFP or isotype control on NSG-PiZ livers 5 h after IV administration of eGFP mRNA-LNP. **d** Representative images of immunofluorescence staining for HGF and EGF on liver tissue of untreated NSG-PiZ mice and 24 h or 48 h after IV administration of HGF + EGF mRNA-LNP. **e** Serum HGF levels measured via ELISA in same mice as **d**. *n* = 3 mice for all time points, except *n* = 4 mice at 72 h. **f** Serum EGF levels measured via ELISA in same mice as **d**. *n* = 3 mice per group. **g** NSG-PiZ mice were injected IV

with either AAV8-TBG-p21 or no vector. One week later mice were treated IV with HGF + EGF mRNA-LNP or control Luc mRNA-LNP. **h** Top row: representative images of EdU⁺ proliferating liver cells after Luc or HGF + EGF injections in group that received no AAV8. Bottom row: representative images of EdU⁺ proliferating liver cells after Luc or HGF + EGF injection in group that was injected with AAV8-TBG-p21. **i** Quantification of EdU⁺ hepatocytes per 10x image field, identified by morphology. *n* = 3 mice per group. At least three images were averaged per mouse, each image from a different liver lobe. Gray−Luc, green−HGF + EGF group. For all panels: each dot represents one mouse, error bars = SEM, scale = 100 μm, *P* values were calculated by unpaired two-sided student's *t* test, ns *P* > 0.05, *\**P* ≤ 0.05, ****P* ≤ 0.01, *\*\*\**P* ≤ 0.001, *\*\*\*\**P* ≤ 0.0001. Source data are provided as a Source Data file. **a, c, d, g** Created with BioRender.com released under a Creative Commons Attribution-NonCommercial-NoDerivs 4.0 International license.

doublet hALB⁺ Ku80⁺ PHHs were detected. In contrast, in the p21 only and HGF + EGF only groups there was a significant increase in the number of PHH clusters (Fig. 2b, c), indicative of PHH survival. Likewise, there was an increase in the number of PHHs per cluster and the number of EdU⁺ PHHs per cluster (Fig. 2d, e), both indicative of PHH proliferation. The combined group showed a significant synergy between the p21 injection and HGF + EGF treatments in improving PHH survival and proliferation (Fig. 2b–e). This was also reflected in the highest hALB⁺ Ku80⁺ area (Fig. 2f), where the combined treatment group had an 18.6-fold increase over the control group. Serum hALB was measured over time to evaluate the kinetic of PHH transplantation and engraftment globally within the livers, to assess PHH function, and to show the trajectory of human hepatocyte expansion (Fig. 2g). Here we show statistically significant differences among all groups at both 1 and 2 weeks post transplantation, with the highest hALB levels achieved in the combined group as expected (27.4-fold increase vs. control group), followed by the HGF + EGF group (12.1-fold increase), and then the p21 group (4.3-fold increase). Together, these findings demonstrate that inducing p21 expression in hepatocytes to mimic halted host hepatocyte proliferation, as seen in human diseased livers, provides a basal level of engraftment advantage to PHHs. Additionally, in this disease context, treatment with HGF + EGF mRNA-LNP drastically synergizes this advantage by significantly increasing the survival and proliferation of PHHs. These findings suggest that the therapeutic intervention of HGF + EGF mRNA-LNP may successfully improve PHH cell therapy in the clinical context of AATD-associated liver disease where p21 is expressed in hepatocytes.

### HGF + EGF mRNA-LNP treatment leads to sustained and robust engraftment of functional PHHs, reduced liver disease burden, and restored liver function in the p21/NSG-PiZ model

Sustained engraftment of functional PHHs over time in the p21/NSG-PiZ mice treated with HGF + EGF mRNA-LNP was further investigated using a similar experimental set up as above. The mRNA-LNP injections were carried out twice weekly for 5 weeks before mice were sacrificed (Fig. 3a). The control group received AAV8-TBG-Null and Luc mRNA-LNP injections, and the experimental group received AAV8-TBG-p21 injection and HGF + EGF mRNA-LNP treatments (Fig. 3a). Here a 5-week treatment scheme was chosen to examine longer-term PHH proliferation, exponential expansion, and engraftment over time in comparison to what has previously been achieved in other PHH transplantation studies[21]. Sustained engraftment is shown in the experimental group by the presence of large clusters of Ku80⁺ PHHs across an entire liver lobe compared to the control in which the clusters remained small and sparse (Fig. 3b). This level of engraftment in the experimental group was consistent across all liver lobes and samples (Supplementary Fig. 3a). This difference was further validated with hALB/Ku80/EdU co-staining (Supplementary Fig. 3b) and was quantified by measuring the percent of liver tissue occupied by hALB⁺ Ku80⁺ PHHs. On average, ~30% liver repopulation was quantified on liver sections in the experimental group as compared to ~2% in the control group (Fig. 3c), yielding a 14.1-fold

increase in the treated group. Importantly, an exponential increase of serum hALB over time was observed in the experimental group with an impressive 37.1-fold increase at 5 weeks compared to the control group (Fig. 3d). Quantification of serum hALB over time is very often used to determine the kinetics of transplantation and engraftment, and as a measure of liver repopulation over time[12–16,18–21]. At this time, p21 expression was observed in 6% of hepatocytes in the control group and 21% in the treated group (Supplementary Fig. 4a, b). This relative decrease in the percentage of p21⁺ hepatocytes as compared to Fig. 1b can be attributed to the fact that a large portion of the livers in the treated group is now occupied by p21⁻ donor PHHs, which are on average smaller than neighboring mouse hepatocytes. The remaining p21⁺ cells are mouse host hepatocytes, as expected.

Five weeks post transplantation, smaller clusters of engrafted Ku80⁺ PHHs were predominantly found in periportal regions, identified by cytokeratin 7 (CK7) expression in bile ducts, and not near pericentral regions, identified with hepatic glutamine synthetase (GS) expression (Supplementary Fig. 5a). This suggests PHHs engraft first in the portal region as expected based on transplantation method. In regions with larger expansion of donor cells, Ku80⁺ PHHs were found in all zones spanning from periportal, mid-lobular, and pericentral regions (Supplementary Fig. 5b–d). In this case, we found that donor PHHs immediately adjacent to central veins express GS, demonstrating that donor cells appropriately adopt expression of the liver zonation markers where they reside (Supplementary Fig. 5d).

Not only did we see high levels of PHH engraftment in the experimental group, but also an improvement in the AATD-associated liver disease phenotype. Indeed, repopulation with healthy hepatocytes significantly decreased the 2C1⁺ misfolded Z-AAT polymer area in the liver as compared to the control transplanted group and age-matched NSG-PiZ controls (Fig. 3e, f). The classic periodic acid Schiff (PAS)-Diastase stain that also highlights the load of Z-AAT globules confirmed this same finding (Supplementary Fig. 6). In line with these data, the experimental group had significantly reduced levels of serum Z-AAT, measured with an ELISA specific for human Z-AAT. This decrease was observed when compared to the control transplanted group as well as the age-matched non-transplanted NSG-PiZ mice, which express only human Z-AAT via the transgene, hence benchmarking the baseline serum human Z-AAT in this mouse strain (Fig. 3g). An ELISA that detects total human AAT, both normal M-AAT and misfolded Z-AAT, also revealed that the experimental group has significantly higher levels of total serum AAT compared to age-matched NSG-PiZ mice (Fig. 3h). The FDA-accepted therapeutic threshold for normal M-AAT protein augmentation is 11 μM, which is equivalent to 572 μg/mL. Taken together, these data show that the engraftment achieved in the experimental group is sufficient to reduce circulating Z-AAT and leads to increased secretion of normal AAT over the age-matched non-transplanted NSG-PiZ mice. As expected, the age-matched NSG mice had no human AAT in either normal or misfolded forms (Fig. 3g, h). As an additional liver function assay we measured ALT levels, where elevated levels above the normal range for NSG mice (>32 IU/L) are indicative of liver damage[40].

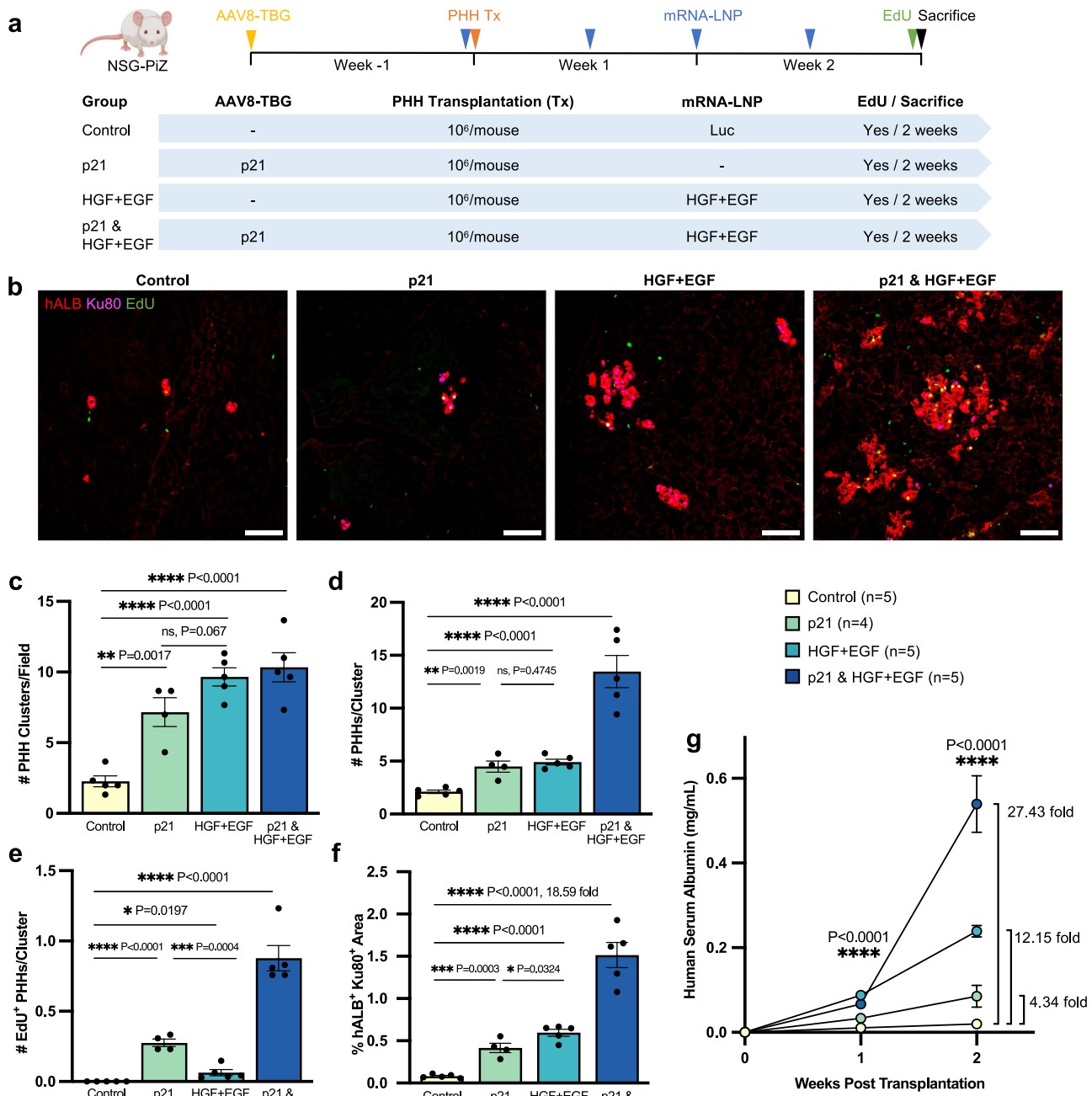

**Fig. 2 | HGF + EGF mRNA-LNP treatments augment PHH engraftment by increasing the survival and proliferation of donor cells in p21/NSG-PiZ mice.** **a** PHH transplantation experimental timeline and treatment groups. A single IV dose of AAV8-TBG-p21 was injected one week prior to cell transplantation, while mRNA-LNP were administered IV 5 h before transplantation and twice weekly for 2 weeks. Cryopreserved PHHs were thawed and transplanted into male NSG-PiZ mice via intrasplenic injection of $10^6$ cells/mouse. Four groups were used: control Luc mRNA-LNP only, AAV8-TBG-p21 only, HGF + EGF mRNA-LNP only, and the combined group with both p21 and HGF + EGF injections. Recipient mice were given an EdU injection and sacrificed 2 weeks post transplantation, serum was collected weekly, and liver tissue harvested for analysis. **b** Representative images of hALB/Ku80/EdU immunofluorescence stain on liver sections 2 weeks post transplantation. **c** Quantification of the number of hALB⁺ Ku80⁺ PHH clusters per 10x image. **d** Quantification of the number of hALB⁺ Ku80⁺ PHHs per cluster. **e** Quantification of the number of hALB⁺ Ku80⁺ EdU⁺ proliferating PHHs per cluster. **f** Quantification of co-stained hALB⁺ Ku80⁺ area per 10x image. For all histology quantification **c**–**f** at least three images were averaged per mouse, each image from a different liver lobe. $P$ values were calculated by unpaired two-sided student's $t$ test. **g** Human serum albumin levels measured via ELISA over the course of 2-week transplantation experiments. $P$ values were calculated by ordinary one-way ANOVA. For all panels: $n = 5$ mice per group except in p21 only group where $n = 4$ mice as specified in figure key, each dot represents one mouse, error bars = SEM, scale = 100 µm, ns $P > 0.05$, *$P \le 0.05$, **$P \le 0.01$, ***$P \le 0.001$, ****$P \le 0.0001$. Yellow−control, green−p21 only, teal−HGF + EGF only, blue−p21 & HGF + EGF treated group. Source data are provided as a Source Data file. **a** Created with BioRender.com released under a Creative Commons Attribution-NonCommercial-NoDerivs 4.0 International license.

ALT levels were significantly lower in the experimental group compared to both the age-matched non-transplanted NSG-PiZ mice as well as the control transplanted group, demonstrating rescue of liver function in the experimental group (Fig. 3i). It is also worth noting here that hepatotoxicity was not observed in these mice after 10 mRNA-LNP injections, confirmed by ALT levels within the expected range in the control and treated transplantation groups (Fig. 3i) and no significant apoptosis detected by TUNEL staining (Supplementary Fig. 7a, b). Together, prolonged treatment with HGF + EGF mRNA-LNP leads to sustained and robust engraftment of functional PHHs, reduces the liver

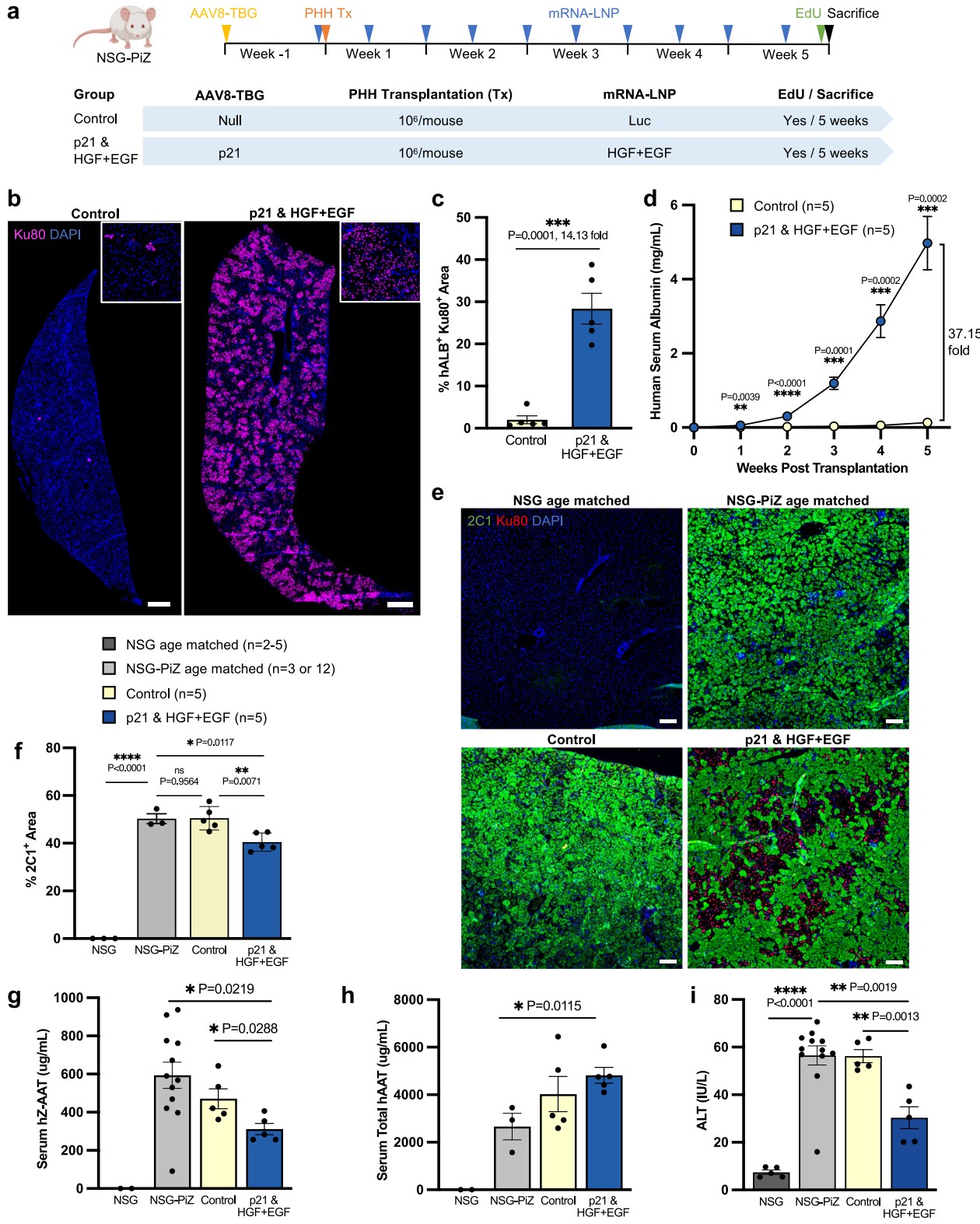

disease burden, and restores overall liver function in the clinically relevant p21/NSG-PiZ model.

### HGF + EGF mRNA-LNP significantly augments PHH engraftment by increasing survival of donor PHHs in the acute p21/APAP overdose model

To expand the translational application of mRNA-LNP with hepatocyte cell therapy to a broader variety of liver diseases, we tested the ability of HGF + EGF mRNA-LNP to improve cell engraftment in the acetaminophen (N-acetyl-para-aminophenol or APAP) overdose model of acute liver injury, which is a leading cause of liver transplantation in the US[41]. This model is well established to induce severe hepatocyte necrosis around the central vein area and resolves within a week following sub-lethal doses[35,42]. Given that p21 expression is naturally occurring in hepatocytes, especially those adjacent to the necrotic areas in the liver of APAP overdosed patients[43], we recapitulated this

**Fig. 3 | HGF + EGF mRNA-LNP treatments lead to sustained and robust engraftment of functional PHHs, which ameliorates liver disease burden and improves overall liver function of p21/NSG-PiZ mice. a** PHH transplantation, AAV8 injections, mRNA-LNP injections, EdU injections, serum collection, and tissue harvesting were carried out as previously described. Two groups were used: control group with AAV8-TBG-Null and Luc mRNA-LNP versus experimental group with both AAV8-TBG-p21 and HGF + EGF mRNA-LNP injections. Mice were sacrificed, and all analyses were performed 5 weeks post transplantation. **b** Representative images of Ku80 immunofluorescence stain on liver sections, insets show detail. Scale = 1000 μm. **c** Quantification of PHH engraftment in mouse liver, measured as percent liver tissue area occupied by co-stained hALB⁺ Ku80⁺ PHHs per 10x image. n = 5 mice per group. **d** Human serum albumin levels measured via ELISA over time. n = 5 mice per group. **e** Representative images of 2C1 immunofluorescence stain on liver sections of control non-transplanted age-matched NSG and NSG-PiZ mice and transplanted groups. Scale = 100 μm. **f** Quantification of % liver tissue area occupied by 2C1⁺ polymers per 10x image. For NSG and NSG-PiZ n = 3 mice, for transplanted groups n = 5 mice. **g** Serum hZ-AAT levels measured via ELISA. This ELISA specifically recognizes misfolded hZ-AAT. For NSG n = 2 mice, for NSG-PiZ n = 12 mice, for both transplanted groups n = 5 mice. **h** Serum hAAT levels measured via ELISA. This ELISA recognizes total hAAT protein, both normal and misfolded. For NSG n = 2 mice, for NSG-PiZ n = 3 mice, for both transplanted groups n = 5 mice. **i** Liver enzyme ALT levels in serum. For NSG n = 5 mice, for NSG-PiZ n = 12 mice, for both transplanted groups n = 5 mice. For all histology quantification **c**, **f** at least 3 images were averaged per mouse, each image from a different liver lobe. For all panels: each dot represents one mouse, error bars = SEM, P values were calculated by unpaired two-sided student's t test, ns P > 0.05, *P ≤ 0.05, **P ≤ 0.01, ***P ≤ 0.001, ****P ≤ 0.0001. Dark gray−NSG, light gray−NSG-PiZ, yellow−control, blue−treated group. Source data are provided as a Source Data file. **a** Created with BioRender.com released under a Creative Commons Attribution-NonCommercial-NoDerivs 4.0 International license.

context by injecting NSG mice with one dose of AAV8-TBG-p21 virus 2 weeks prior to APAP overdose. Then, APAP was injected to mimic acute sub-lethal overdose[35,44], and PHHs were transplanted 24 h later. mRNA-LNP were administered 5 h before transplantation and twice weekly thereafter for 2 weeks (Fig. 4a). The control group was given AAV8-TBG-Null and Luc mRNA-LNP injections, while the experimental group was given one AAV8-TBG-p21 injection and HGF + EGF mRNA-LNP treatments (Fig. 4a). P21 expression was validated in the majority of hepatocytes in the experimental group (Fig. 4b). Liver histological analyses 2 weeks after cell transplantation revealed that clusters of PHHs were mainly singlet and doublet hALB⁺ Ku80⁺ cells and were visible in both groups, yet the number of clusters was significantly greater in the experimental group, indicating that HGF + EGF mRNA-LNP treatment promotes PHH survival in the p21/APAP model (Fig. 4c−e). However, PHHs within the clusters were no longer proliferative 2 weeks post transplantation, likely because the injury had naturally recovered by that time (Fig. 4f). In line with the more abundant PHH clusters in the experimental group, the percentage of hALB⁺ Ku80⁺ positive area was significantly higher as compared to the control group (Fig. 4g). This result was further validated by a significant increase in serum hALB levels in the experimental group 1 week post transplantation, reflecting global PHH engraftment within entire livers (Fig. 4h). Together, these findings expand the therapeutic indications of HGF + EGF mRNA-LNP for liver cell therapy from chronic to acute liver diseases. We specifically demonstrate that HGF + EGF mRNA-LNP treatments improve the survival and, thus, engraftment of transplanted PHHs in a clinically relevant p21/APAP acute liver injury model.

## Discussion

PHH cell therapy offers advantages over liver transplantation, the current standard of care for end stage liver disease. Donor organ shortage could be addressed with PHH transplantation, as one donor organ could treat multiple patients and offers the convenience of cryopreservation for future use. PHH transplantation is less invasive than liver transplantation and can be repeated. In cases of acute liver injury, PHHs could provide temporary support while the host liver naturally regenerates, bridging the gap to liver transplantation if needed. Although, clinical trials using PHH transplantation to treat liver diseases have shown the safety of the procedure, cell engraftment remains poor. In this study, we explore the potential of mitogenic HGF + EGF mRNA-LNP to improve PHH engraftment to potentiate a viable cell therapy for both chronic and acute liver injuries.

Previous PHH transplantation mouse models achieve significant engraftment levels but require a prolonged 5 to 9-month period post transplantation and harsh preconditioning techniques that are not plausible in a clinical setting. Those approaches included liver irradiation, partial hepatectomy, monocrotaline, anti-CD95 antibodies, induced acute injury, and urokinase-type plasminogen activator transgenes or adenovectors[12–22,45,46]. Such strategies either create

physical room for transplanted cells, inhibit host liver proliferation, or damage the liver parenchyma to allow donor cells to engraft. In addition, regenerative stimuli for PHH engraftment have been induced via portal vein occlusion or partial hepatectomy[5,10,21]. Although PHH engraftment was successful to varying degrees in these studies, the approaches utilized are not clinically translatable nor represent accurate models of human liver diseases.

In the present work, our focus was on creating a clinically relevant setting to achieve better PHH engraftment in a shorter period to treat a broad range of liver diseases. We propose that the naturally occurring hepatocyte senescence observed in virtually all human liver diseases[32,33,43] provides a growth advantage for transplanted PHHs over the host hepatocytes. This is a diseased context that we mimicked in our experimental models by inducing AAV-mediated p21 expression in hepatocytes. Our approach also relies on controlled delivery of hepatocyte regenerative stimuli, HGF and EGF, to the liver using mRNA-LNP[23,24], which has been validated as clinically safe with the recent mRNA-based COVID-19 vaccines. This is an advantageous method of delivery that we pioneered in this study to improve cell therapy, as it leads to robust yet controlled expression of mitogens in the liver[23,24]. The HGF/cMET axis is an appealing target for improving engraftment, as supported by other cell therapy applications using HGF-expressing adenovectors or cMET agonist antibodies for primary mouse and stem cell-derived hepatocytes, respectively[47,48]. Synergy between the HGF/cMET and EGF/EGFR axes during liver development and regeneration has been reported by many groups[28–31] including ours[23], therefore, a combination of both HGF and EGF is used in the present study.

Most of the previous PHH therapy studies analyzed cell engraftment months after transplantation, which does not provide understanding of the mechanism of initial cell repopulation. As a major limitation to successful engraftment is poor cell survival after cell injection, here, we intentionally examine PHH engraftment and function during the first 2 weeks after transplantation. We evaluated the potential of HGF + EGF mRNA-LNP to improve not only cell proliferation but also survival, as this also dictates the rate at which cell repopulation occurs. Our study demonstrates that HGF + EGF mRNA-LNP treatments lead to significantly improved PHH survival upon transplantation, increased proliferation over time, and sustained PHH engraftment for at least 5 weeks in the chronic p21/NSG-PiZ model. Engrafted PHHs contribute to a large percentage of overall liver area, an impressive and clinically relevant 30%. The engrafted cells harbor normal hepatocyte functions, notably secreting high levels of hALB and hAAT above the FDA-accepted protein augmentation therapeutic threshold[21]. Engraftment of these functional cells reduces misfolded Z-AAT protein burden in liver tissue and sera, and ALT levels are significantly reduced indicating restoration of overall liver function. There is a growing body of evidence that demonstrates a correlation between circulating Z-AAT polymers and overall liver disease in AATD patients, thus, reducing serum Z-AAT is promising for overall patient

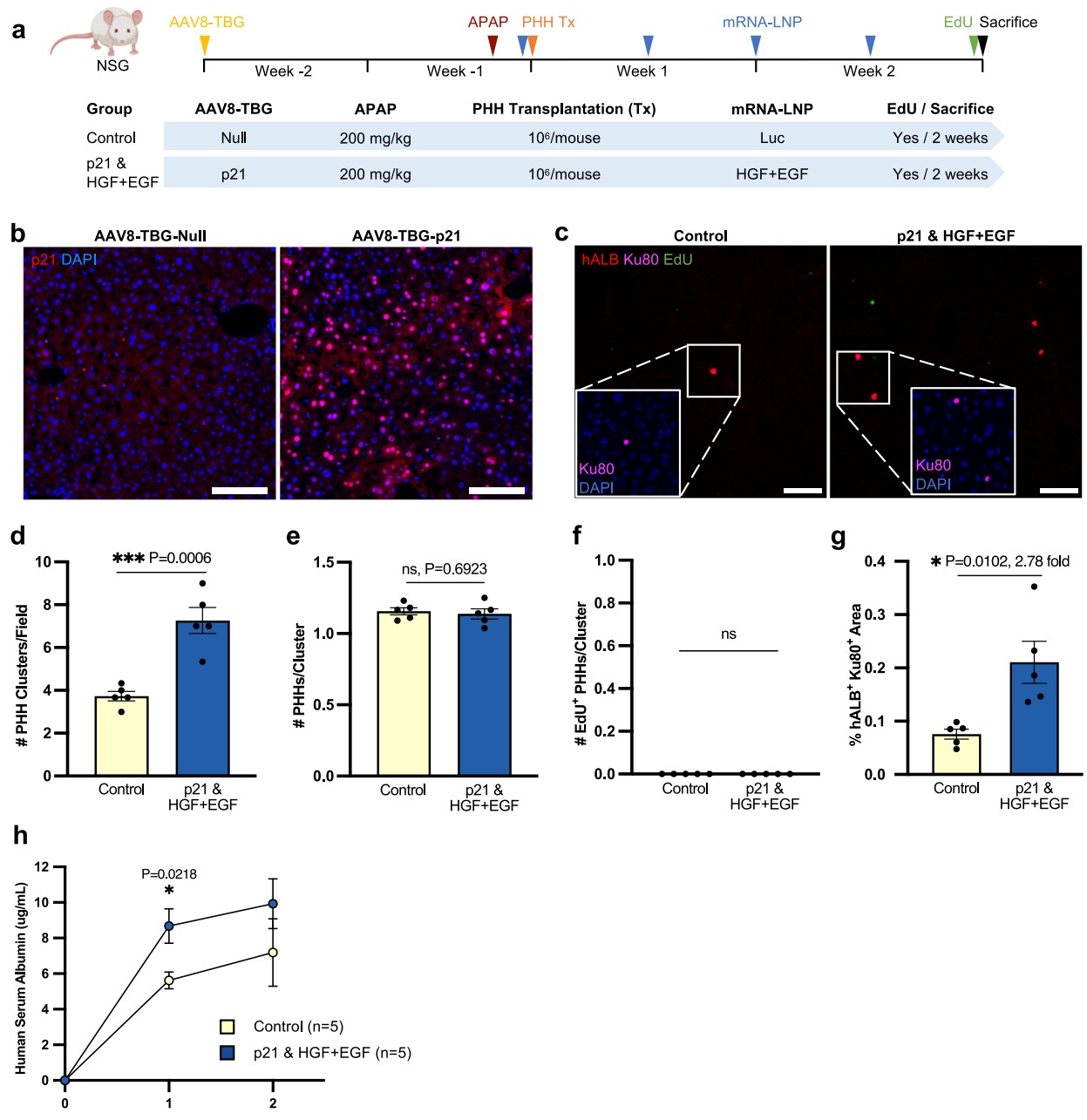

**Fig. 4 | HGF + EGF mRNA-LNP treatment augments PHH engraftment by increasing survival of donor cells in acute p21/APAP overdose model. a** PHH transplantation experimental timeline and treatment groups. A single IV dose of AAV8-TBG-Null or AAV8-TBG-p21 was injected two weeks prior to cell transplantation. APAP was administered IP at 200 mg/kg. Cryopreserved PHHs were thawed and transplanted into male NSG mice via intrasplenic injection of $10^6$ cells/mouse 24 h later. mRNA-LNP were administered IV 5 h before transplantation and twice weekly for two weeks. Two groups were used: the control group with AAV8-TBG-Null and Luc mRNA-LNP versus the experimental group with both AAV8-TBG-p21 and HGF + EGF mRNA-LNP injections. Recipient mice were given an EdU injection and sacrificed 2 weeks post transplantation, serum was collected weekly, and liver tissue harvested for analysis. **b** Representative images of p21 immunofluorescence stain. **c** Representative images of hALB/Ku80/EdU immunofluorescence stain on liver sections 2 weeks post transplantation. Insets are zoomed in images to show detail. **d** Quantification of the number of hALB$^+$ Ku80$^+$ PHH clusters per 10x histology image. **e** Quantification of the number of hALB$^+$ Ku80$^+$ PHHs per cluster. **f** Quantification of the number of hALB$^+$ Ku80$^+$ EdU$^+$ proliferating PHHs per cluster. **g** Quantification of co-stained hALB$^+$ Ku80$^+$ area per 10x image. **h** Human serum albumin levels measured via ELISA over the course of 2-week transplantation experiments. For all histology quantification **d–g** at least 3 images were averaged per mouse, each image from a different liver lobe. For all panels: $n = 5$ mice per group, each dot represents one mouse, error bars = SEM, scale = 100 μm, $P$ values were calculated by unpaired two-sided student's $t$ test, ns $P > 0.05$, * $P \le 0.05$, **$P \le 0.01$, ***$P \le 0.001$, ****$P \le 0.0001$. Yellow−control, blue−treated group. Source data are provided as a Source Data file. **a** Created with BioRender.com released under a Creative Commons Attribution-NonCommercial-NoDerivs 4.0 International license.

outcomes[49,50]. Approximately 3.4 million individuals worldwide carry disease alleles for AATD, with no existing cure for associated liver disease other than liver transplantation[51]. If the results of our study are translated to human, successful PHH cell therapy offers a potential cure for AATD patients. This cell therapy holds the promise of addressing both liver and lung diseases associated with AATD by ameliorating the gain of function toxicity of Z-AAT in the liver via replacing diseased cells and addressing the loss of function in the lung via

restoring normal AAT protein in the serum. We also expanded the clinical application of our approach to possibly accelerate liver regeneration after APAP overdose acute liver injury. Our data reveal that HGF + EGF mRNA-LNP treatment significantly enhances PHH survival, engraftment, and function during the first two weeks after transplantation in a sub-lethal APAP overdose model. Given that APAP overdose is one of the leading causes of liver transplantation in the US[41], there is a critical need for alternative therapies for patients who arrive at the emergency room too late to be efficiently treated with the standard of care, N-acetyl cysteine. If our results are translated to human acute liver injury cases, improved PHH therapy with HGF + EGF mRNA-LNP could serve as a bridge to provide time for the host's natural liver repair process to take over or ultimately for whole organ transplantation.

Overall, our study pioneers the use of the nucleoside-modified mRNA-LNP platform to express mitogens in the liver in a controlled manner to drastically improve PHH survival, proliferation, and engraftment, thus, reducing disease burden and enhancing overall liver function. This strategy may overcome the critical barriers to clinical translation of hepatocyte cell therapies, primary or stem cell-derived, for the treatment of various liver diseases.

## Methods

### Chronic and acute liver injury mouse models
All animal studies were approved by the Boston University IACUC and were consistent with local, state, and federal regulations as applicable. Animals were housed under standard conditions with a 12-hour day/night cycle in a pathogen-free environment with access to food and water ad libitum. All mice used in Fig. 1 were male and ranged from 7 to 10 weeks old. All mice used for xenotransplantation studies in Figs. 2–4 were male and 7–8 weeks old at the time of transplantation. For chronic injury, the transgenic NOD.Cg-Prkdc[scid] Il2rg[tm1Wjl] Tg(SERPINA1*E342K)#Slcw/SzJ (NSG-PiZ) colony was used, mimicking human AATD-associated liver disease. NSG-PiZ mice are homozygous for a transgene expressing the human mutated PiZ allele. For acute liver injury, immunodeficient NOD.Cg-Prkdc[scid] Il2rg[tm1Wjl]/SzJ (NSG) mice were used. Following a 14 h fast to induce consistent liver damage among the mice, liver injury was induced with a single intraperitoneal injection of acetaminophen (N-acetyl-para-aminophenol, paracetamol, APAP) at 200 mg/kg diluted in sterile PBS (Spectrum Chemical, AC100). Mice were maintained on a normal chow diet and water ad libitum after APAP injections. Both mouse strains were obtained from The Jackson Laboratory (JAX stock #028842 and #005557). As soon as NSG or NSG-PiZ mice were to be used for cell transplantation, they were treated with sulfamethoxazole/trimethoprim (Sulfatrim) in sterile drinking water on alternate weeks as previously described[52]. This antibiotic-supplemented water was given at a dose of 0.6720 mg/mL drinking solution, protected from light. The water was changed, and fresh antibiotic-supplemented water was given every three days on alternate weeks.

### In vivo administration of AAV8-TBG-p21 and AAV8-TBG-Null
The AAV8.TBG.PI.p21.WPRE.bGH vector was obtained from the Penn Vector Core. The AAV8.TBG.PI.Null.bGH vector was obtained from Addgene (105536-AAV8). Mice were anesthetized with isoflurane, and the vectors were thawed, diluted in sterile PBS, and injected retro-orbitally at a dose of $5.00 \times 10^{11}$ gc/mouse using 29 G Exel International Insulin Syringes (0.5 mL).

### PHH source and transplantation
All the following protocols were carried out using aseptic technique in ABSL-2 hoods and using sterile supplies. Cryopreserved PHHs (BioIVT, Lot DJW) were stored at 150 °C, thawed, and added immediately to 50 mL warm INVITROGRO CP Medium (BioIVT Z990003). The tube was inverted three times slowly to resuspend cells. The cell suspension

was centrifuged at $50 \times g$ for 5 min at room temperature. The supernatant was discarded, and PHHs were resuspended in 5 mL of Hank's Balanced Salt Solution (HBSS, Gibco 14175095), counted using trypan blue method, and centrifuged again. The supernatant was removed, and cells were resuspended and aliquoted to administer 1 million cells per mouse in ~50 µL HBSS. Cells were kept on ice and immediately transplanted via intrasplenic injection as previously described[53]. Mice were anesthetized with a ketamine/xylazine cocktail via intraperitoneal injection. BuprenorphineSR painkiller (0.5–1.0 mg/kg) was administered subcutaneously to provide 72 h of analgesia. Ophthalmic ointment was applied to the eyes using the sterile cotton-tipped applicator. Fur was shaved with an electric razor around the spleen area on the left side. Disinfection of the surgical site was performed with iodine pads (Dynarex™ 1108) followed by alcohol wipes, repeated three times. Mice were kept warm with a heating pad under the tail and transferred to the sterile field. Sterile drapes, gloves, and surgical instruments were used. An incision of 1 cm was made with a small scissor on the mouse's left paralumbar fossa to expose the spleen out of the body. The spleen was kept moist with PBS. One million PHHs resuspended in 50 µL of HBSS were injected into the inferior pole of the spleen using a 27 G tuberculin syringe with an attached needle (BD 305620). Surgifoam gelatin sponge (Ethicon) was used to achieve hemostasis by pressuring a small piece on the injection site for 30 s. The exposed spleen was quickly returned to the abdominal cavity. The body wall was closed with an absorbable suture (Med Vet International VR494), and the skin was closed with 2 wound clips (Reflex, 7 mm). Mice were observed post anesthesia until they awoke.

### mRNA production
mRNAs were produced using T7 RNA polymerase (Megascript, Ambion) on linearized plasmids encoding codon-optimized firefly luciferase (Luc)[54,55], eGFP, HGF, and EGF. mRNAs were transcribed to contain 101 nucleotide-long poly(A) tails. One-methylpseudouridine (m1Ψ)−5'-triphosphate (TriLink) instead of UTP was used to generate modified nucleoside-containing mRNA. RNAs were capped using the m7G capping kit with 2'-O-methyltransferase (ScriptCap, CellScript) to obtain cap1. mRNAs were purified by cellulose-based purification as described[56]. All mRNAs were analyzed by agarose gel electrophoresis and were stored frozen at −20 °C. The optimized coding sequences of DNA for making nucleoside-modified mRNA are listed in Supplemental Table 1.

### LNP formulation of the mRNA
FPLC-purified m1Ψ- containing mRNAs were encapsulated in LNP using a self-assembly process in which an aqueous solution of mRNA at pH 4.0 was rapidly mixed with a solution of lipids dissolved in ethanol[57]. LNP used in this study contains an ionizable cationic lipid (pKa in the range of 6.0–6.5, proprietary to Acuitas Therapeutics)/distearoylphosphatidylcholine/cholesterol/PEG-lipid[57,58]. The proprietary lipid and LNP composition are described in US patent US10,221,127 entitled "Lipids and lipid nanoparticle formulations for delivery of nucleic acids" (https://www.lens.org/lens/ patent/183-348-727-217-109)[59]. They had a diameter of ~80 nm as measured by dynamic light scattering using a Zetasizer Nano ZS (Malvern Instruments Ltd, Malvern, UK) instrument. Acuitas will provide the LNP used in this work to academic investigators who would like to test it.

### In vivo administration of mRNA-LNP
mRNA-LNP were thawed and freshly diluted on ice in sterile Dulbecco's phosphate-buffered saline (DPBS) prior to each experiment. Mice were anesthetized with isoflurane and administered 50 µL of diluted mRNA-LNP encoding luciferase (10 µg/mouse), eGFP (10 µg/mouse), or combination HGF (5 µg/mouse) + EGF (5 µg/mouse) intravenously by retro-orbital injection using 29 G Exel International Insulin Syringes (0.5 mL).

### EdU injection, animal sacrifice, serum collection, liver tissue fixation, cryopreservation, and sectioning

2 h prior to sacrifice, mice were injected intraperitoneally with 200 µL of EdU solution at 5 mg/mL in sterile DPBS (Sigma Aldrich 90058450MG). Euthanasia was performed at indicated end points by $CO_2$ inhalation, after which death was determined by cessation of breathing, lack of pulse, and limb pinching. Then, cervical dislocation was performed. This method of euthanasia is consistent with the recommendations of the American Veterinary Medical Association (AVMA) Guidelines for the Euthanasia of Animals. Whole blood samples were collected from the inferior vena cava and allowed to clot at room temperature for 15 min. Then, blood samples were centrifuged at $1000–2000 \times g$ for 15 min at room temperature. The serum layer was separated and kept at −80 °C until required for analysis. Liver and spleen tissues were harvested and fixed in 4% paraformaldehyde (Electron Microscopy Sciences 15710) solution overnight at 4 °C. The next day, the tissues were placed in a 15% sucrose solution for 30 min, and then placed in a 30% sucrose solution for at least 48 h. Fixed liver and spleen tissues were flash frozen into OCT (Fisher 23-730-571) blocks and sectioned using a cryostat (CM1950 Leica) at 5 µm thickness onto microscope slides (Fisher 12-550-15). Sections were stored at −20 °C until required for immunostaining.

### Immunohistochemistry on PFA-fixed frozen tissue sections

The frozen sections were allowed to defrost and dry at room temperature for 10 min. The slides were dipped in PBS for 10 min, then permeabilized using 0.3% triton X in PBS for 10 min. The slides were rinsed three times in PBS, 10 min each, and blocked with 3% normal donkey serum for 30 min. The sections were then incubated overnight at 4 °C with appropriate primary antibodies or antibodies diluted in PBS with 0.1% BSA at concentrations indicated in Supplemental Table 2. For nuclear staining with p21 and Ku80, 0.2% triton X was also included in the overnight primary antibody incubation. For EGF and 2C1 staining, the Mouse on Mouse kit (Vector Labs, BMK-2202) was used to reduce background from mouse tissue. Following primary antibody incubations, the slides were washed three times with PBS, 10 min each. Then slides were incubated with the appropriate fluorescent labeled secondary antibodies for 1 h at room temperature, protected from light. The slides were washed three times in PBS for 10 min each. Finally, the slides were incubated with DAPI (1:3000 in PBS) for 3 min, washed three times in PBS, and mounted using FluorSave reagent (MilliporeSigma, 345789-20 mL) and a coverslip (Epredia, 102450). For EdU detection, staining was performed using Click-iT EdU Imaging Kit (Invitrogen C10337) after the secondary antibody incubation step following the manufacturer's protocol. The Invitrogen Click-iT Plus TUNEL Assay Kit for In Situ Apoptosis Detection (Invitrogen, C10617) was used to stain apoptotic and necrotic cells.

### PAS-Diastase stain

Periodic acid Schiff (PAS) staining stains both glycogen and Z-AAT globules. Diastase digests away glycogen, allowing for visualization of only misfolded Z-AAT protein. First, PFA-fixed frozen tissue sections were brought to room temperature and hydrated with diH2O for 5 min. The diastase/a-amylase solution was prepared fresh by adding 0.4 g diastase (Sigma Aldrich, A3176) to 80 mL diH2O, microwaving for 30 s, and mixing gently. Tissue sections were placed in the warmed diastase solution for 45 min. Slides were washed in several changes of diH2O, followed by PAS staining using a kit (Sigma Aldrich, 395B). Sections were oxidized with a periodic acid solution for 5 min, rinsed in several changes of diH2O, and incubated in Schiff's Reagent for 15 min. After rinsing in diH2O again, sections were dehydrated and mounted with permanent mounting media.

### Microscopy, image analysis, and quantification

All images were taken using a Nikon microscope (Nikon Eclipse Ni-E) and analyzed using NIS Elements software and Fiji. For all histology quantification, measurements for each mouse were averaged using three 10× images per mouse, each from a different liver lobe. For quantifying EdU⁺ or p21⁺hepatocytes, positive cells were counted manually based on morphology using high power magnification. To quantify the number of PHH clusters/field, cells/cluster, and EdU⁺ cells/cluster, positive events were manually counted per image using high-power magnification. To quantify the percent hALB⁺ or 2C1⁺ area, positive areas were measured using Fiji and were divided by the total tissue area, accounting for empty areas due to large vasculature such as portal and central veins. The percentage of liver repopulation was quantified by tracing the area occupied by hALB⁺ Ku80⁺ PHHs and dividing by total tissue area, accounting for empty areas due to large vasculature such as portal and central veins.

### Enzyme-linked immunosorbent assays (ELISA)

Serum samples were stored at −80 °C, thawed on ice, and used immediately. Repeat freeze-thaw cycles were avoided. The following ELISAs were performed according to the manufacturer's protocols, specifically following the instructions for serum samples: human albumin (Bethyl Laboratories, E88-129), human HGF (Abcam, ab275901), and human EGF (Abcam, ab217772). The total human AAT ELISA was performed using a quantification kit (GenWay Biotech). This protocol was then adapted to include a Z-AAT antibody and standard as described here[60].

### ALT assay

Assays were performed using the Pointe Scientific kit (A7526-450) for testing serum ALT levels following manufacturer's protocol. Briefly, 10 µL of serum was mixed with supplied reagent mix at 37 °C and readings were measured at 340 nm every 1 min for 5 min using Molecular Devices SpectraMax i3x Multi-Mode microplate reader.

### Figure generation

Cartoons and schematics were created with BioRender.com released under a Creative Commons Attribution-NonCommercial-NoDerivs 4.0 International license.

### Statistics and reproducibility

All statistical analyses were performed using GraphPad Prism version 10.0.0 for Mac, GraphPad Software, Boston, Massachusetts USA, www.graphpad.com. For comparison between the two groups, an unpaired two-sided $t$ test was used to calculate statistical significance. For comparing multiple groups, an ordinary one-way unpaired ANOVA was performed. Quantitative data are shown as mean ± standard error of the mean (SEM) and are considered statistically significant when $p < 0.05$ (ns $P > 0.05$, *$P \le 0.05$, **$P \le 0.01$, ***$P \le 0.001$, ****$P \le 0.0001$).

### Reporting summary

Further information on research design is available in the Nature Portfolio Reporting Summary linked to this article.

## Data availability

All data generated and analyzed during this study are included in this published article and its Supplementary Information. Sequences are included in Supplementary Information. Source data are provided with this paper.

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

## Acknowledgements

This work is supported by: NIH NIDDK R01DK124361-01A1, NIH NIDDK 1F31DK135378-01, Boston University CTSI TL1 Pre-Doctoral Fellowship in Regenerative Medicine TL1TR001410, March of Dimes Research Grant #6-FY14-530, and Alpha-1 Foundation Research Grants #614163 and #640084. We would like to acknowledge Drs. Greg Miller and Marianne James of the CReM, supported by grants R24HL123828 and U01TR001810. We also acknowledge Samuel Morningstar and Dilnar Mahmut, M.S., for proofreading this manuscript. We would also like to recognize Drs. Andrew Wilson and Joseph Kaserman for their technical assistance with the AAT and Z-AAT ELISAs. This work was supported by the Boston University Analytical Instrumentation Core and the Boston University Animal Science Center.

## Author contributions

A.R.S., F.R., E.E., N.P., D.W., and V.G.-E. conceived and designed experiments. A.R.S., A.A., S.W., and V.G.-E. performed experiments. H.M., N.P., and D.W. designed and prepared the mRNA. Y.T. designed and prepared the LNP. A.R.S.and V.G.-E. wrote the manuscript. V.G.-E. directed the research.

## Competing interests

Declaration of interests in accordance with the University of Pennsylvania's policies and procedures and our ethical obligations as researchers, we report that D.W. is named on patents that describe the use of nucleoside-modified mRNA as a platform to deliver therapeutic proteins. D.W. and N.P. are also named on a patent describing the use of modified mRNA in lipid nanoparticles. D.W., N.P., and V.G.-E. are named on a patent describing the use of nucleoside-modified mRNA in lipid nanoparticles to treat liver diseases. D.W. has a provisional international patent application that describes the use of nucleoside-modified mRNA in lipid nanoparticles to treat liver diseases. D.W. is a co-founder and has a financial interest in Pittsburgh ReLiver, Inc. Y.T. is an employee of Acuitas Therapeutics, a company focused on the development of lipid nanoparticulate nucleic acid delivery systems for therapeutic applications. All other authors declare no competing interests.
