## [Peer Review File · Nature Communications]

Transient growth factor expression via mRNA in lipid nanoparticles promotes hepatocyte cell therapy in miceReviewers' Comments:

Reviewer #1:

Remarks to the Author:

Authors did a good job explaining the need of the proposed treatment. The design and implementation of an alternative animal model shows a more translatable approach into humans. Additionally, authors should clarify whether these results would be sufficient enough for treatment of these diseases or if further improvements are needed in order for readers to determine significance of the results in their current form. Finally, authors also show a good demonstration of potential use of this treatment for acute liver diseases as well as chronic. With the small adjustments mentioned below, the paper is ready for publication.

Specific Comments

1. Page 4, Figure 1- Authors mention that circulating HGF and EGF peak at 24 hours and decreases at 48 hours, however these are the only time points shown. Are there more time points analyzed to make the accurate statement that 24 hours was the peak? Otherwise, using "peaked" may not be an accurate description as it is not known if this is a true peak or the peak is between 0-24 or 24-48.
2. Page 4 figure 2/Page 5 figure 3- Some explanation of why this timeline was chosen would allow for a more complete story
3. Page 5 line 78- Says that PHHs were shown across entire liver lobe, should be clarified if other lobes were analyzed as well or if this is representative of all liver lobes

Reviewer #2:

Remarks to the Author:

The authors provide detail information related to improved engraftment and survival of primary human hepatocytes (PHH) in a mouse model of alpha antitrypsin deficiency. p21 is administered to mimic the accumulation of senescent hepatocytes seen in most human liver diseases. The authors conclusively demonstrate that HGF and EGF mRNAs administered through lipid nanoparticles substantially increased survival and function of the engrafted human hepatocytes as assessed by circulating human albumin. The data are well presented for the reader. The authors should comment in the Discussion on the following:

1. Are the engrafted cells predominantly in a periportal or pericentral position or equally in both locations?
2. Were there any ductules of human origin seen in the engrafted PHH sites?
3. Figure 3B shows that the percent of hepatocytes in the successful engraftment are a very large percent of the overall population of mouse and human hepatocytes. The authors should emphasize this in the Discussion.

Reviewer #3:

Remarks to the Author:

Major revision

The article demonstrates the potential of AAV-mediated p21 expression and mitogenic HGF+EGF mRNA-LNP to improve PHH engraftment in two mouse models of human chronic and acute liver diseases, especially in the alpha-1 antitrypsin deficiency (AATD) model. Overall, this is a new strategy for human hepatocyte transplantation. However, there are several issues that need to be further addressed.

Major points:

1, Previous studies (PMID: 21505264; PMID: 29032169) demonstrated significant hepatocyte repopulation in NSG-PiZ mice, even without proliferative stimulation treatment. It's thus important to determine if the findings in this paper are model-specific. Have similar results been observed in other chronic models?

2, Regarding the delivery of p21, the authors are encouraged to explain in details the delivery method, duration of action, and its effects. How much p21 is considered effective? And how long these cells should stay p21 positive? Given that many cells are already p21 positive in the disease livers, it is important to determine the optimal dosage and duration of p21 delivery for therapeutic efficacy.

3, The authors should analyze the kinetics of transplantation efficiency in this system, including engraftment and repopulation.

4, For the liver delivery of LNP, please provide detailed information regarding the dosage used, as well as the pharmacokinetics (PK) and pharmacodynamics (PD) of the LNP-mediated delivery. Additionally, it is crucial to include data and analysis related to potential hepatotoxicity associated with the LNP treatment.

5, This liver delivery strategy is clinically significant, and it is important to assess the consistency of LNP delivery efficiency in human hepatocytes, such as in a hepatocyte-humanized animal model.

6, APAP is an acute liver injury model. The authors are encouraged to perform cell transplantation in a clinically related chronic liver disease model.

Minor points:

1, Absence of control group (Fig. 1a).

2, Lack of NSG age matched and NSG-PiZ age matched groups (Fig. 3b-f).

3, The indicators lack a standard range (Fig. 3g-i).

Author's point-by-point response (blue) to the reviewers' comments, reproduced verbatim (black)

We greatly appreciate the comments from the three reviewers to help us improve the quality of the manuscript as well as the editor's decision to consider our revised manuscript. Here is our point-by-point response to the reviewers' comments. The main manuscript and supplementary information have been edited accordingly. Changes from the original submission are highlighted in blue in the accompanying manuscript file.

Reviewer #1 (Remarks to the Author):

Authors did a good job explaining the need of the proposed treatment. The design and implementation of an alternative animal model shows a more translatable approach into humans. Additionally, authors should clarify whether these results would be sufficient enough for treatment of these diseases or if further improvements are needed in order for readers to determine significance of the results in their current form. Finally, authors also show a good demonstration of potential use of this treatment for acute liver diseases as well as chronic. With the small adjustments mentioned below, the paper is ready for publication.

We thank Reviewer #1 for their insightful and enthusiastic comments. We have further clarified in the Discussion regarding this point and elaborate whether our results would be sufficient for treatment of these diseases. To summarize:

Regarding chronic AATD model: "If the results of our study are translated to human, successful PHH cell therapy offers a potential cure for AATD patients. This cell therapy holds the promise of addressing both liver and lung diseases associated with AATD by ameliorating the gain of function toxicity of Z-AAT in the liver via replacing diseased cells and addressing the loss of function in the lung via restoring normal AAT protein in the serum." Please see page 7 last paragraph.

Regarding acute APAP model: "If our results are translated to human acute liver injury cases, improved PHH therapy with HGF+EGF mRNA-LNP could serve as a bridge to provide time for the natural liver repair process to take over or ultimately for whole organ transplantation." Please see page 8 first paragraph.

Specific Comments

1. Page 4, Figure 1- Authors mention that circulating HGF and EGF peak at 24 hours and decreases at 48 hours, however these are the only time points shown. Are there more time points analyzed to make the accurate statement that 24 hours was the peak? Otherwise, using "peaked" may not be an accurate description as it is not known if this is a true peak or the peak is between 0-24 or 24-48.

We addressed this comment by including additional time points at 5 hours and 72 hours post injection of HGF+EGF mRNA-LNP. The results have been added to Fig. 1e and Fig. 1f. These results confirm that serum HGF and EGF levels peak at 24 hours post injection. We have also previously published similar results from our lab, thoroughly demonstrating the kinetics of mRNA-LNP mediated protein expression in the murine liver (PMID: 33504774, PMID: 34722830).

The Results section of our manuscript now includes, "The kinetic of expression of mitogens HGF and EGF were then evaluated following co-injection of HGF+EGF mRNA-LNP and analyzed 5, 24, 48, and 72 hours later. HGF and EGF proteins were abundantly detected in the liver tissue of treated mice, while absent in untreated mice at 0 hours (Fig. 1d, Supplementary Fig. 1b, c). Similarly, as HGF and EGF are secreted proteins, circulating serum HGF and EGF peaked at 24 hours after administration and decreased by 48 and 72 hours (Fig. 1e, f). This general kinetic of mRNA-LNP induced protein expression in the mouse liver has also been established and validated previously by our lab^{23,24}." Please see page 4 first paragraph.

2. Page 4 figure 2/Page 5 figure 3- Some explanation of why this timeline was chosen would allow for a more complete story

We have addressed this point by including the following explanation in the main text of the manuscript.

For Figure 2: “We chose to analyze cell engraftment during the first 2 weeks post transplantation, early time points that are often neglected in other studies. This helps to specifically examine the benefit of HGF+EGF mRNA-LNP not only on cell proliferation over a short time, but also on cell survival upon initial transplantation; indeed, improving cell survival is key for efficient and successful cell therapy.” Please see page 5 first paragraph.

For Figure 3: “Here, a 5 week treatment scheme was chosen to examine longer term PHH proliferation, exponential expansion, and engraftment over time in comparison to what has previously been achieved in other PHH transplantation studies.²¹” Please see page 5 second paragraph.

3. Page 5 line 78- Says that PHHs were shown across entire liver lobe, should be clarified if other lobes were analyzed as well or if this is representative of all liver lobes

We agree with this comment and have now included additional representative images of Ku80 immunofluorescence stain on liver sections from the experimental group 5 weeks post transplantation in Supplementary Fig. 2a. One representative lobe is shown from each of the five samples, and various liver lobes are highlighted. Accordingly, we included description of these new data in the Results of our manuscript: “Sustained engraftment is shown in the experimental group by the presence of large clusters of Ku80⁺ PHHs across an entire liver lobe compared to the control in which the clusters remain small and sparse (Fig. 3b). This level of engraftment in the experimental group was consistent across all liver lobes and samples (Supplementary Fig. 2a).” Please see page 5 second paragraph.

Furthermore, the Methods sections details how we quantify all our histology using at least 3 images from 3 unique liver lobes per sample to capture any lobe-to-lobe variability if there is any. This text in the Methods sections is as follows: “For all histology quantification, measurements for each mouse were averaged using three 10X images per mouse each from a different liver lobe.” Please see page 14 second paragraph.

Reviewer #2 (Remarks to the Author):

The authors provide detail information related to improved engraftment and survival of primary human hepatocytes (PHH) in a mouse model of alpha antitrypsin deficiency. p21 is administered to mimic the accumulation of senescent hepatocytes see in most human liver diseases. The authors conclusively demonstrate that HGF and EGF mRNAs administered through lipid nanoparticles substantially increased survival and function of the engrafted human hepatocytes as assessed by circulating human albumin. The data are well presented for the reader. The authors should comment in the Discussion on the following:

We thank Reviewer #2 for their interesting and valuable feedback.

1. Are the engrafted cells predominantly in a periportal or pericentral position or equally in both locations?

This is an interesting point that we addressed by performing immunofluorescence staining on serial sections with well-known portal vein region and central vein region markers. The results are now included as Supplementary Fig. 3 and the Results section texts reads, "Five weeks post transplantation, smaller clusters of engrafted Ku80⁺ PHHs were predominantly found in periportal regions as determined by cytokeratin 7 (CK7) expression in bile ducts and not near pericentral regions identified with hepatic glutamine synthetase (GS) expression (Supplementary Fig. 3a). This suggests they originate in the portal region as expected based on transplantation method. In regions with larger expansion of donor cells, Ku80⁺ PHHs were found in all zones spanning from periportal, mid-lobular, and pericentral regions (Supplementary Fig. 3b, c, d). In this case, we found that donor PHHs immediately adjacent to central veins express GS, demonstrating that donor cells appropriately adopt expression of the liver zonation markers where they reside (Supplementary Fig. 3d)." Please see page 5 last paragraph and page 6 first paragraph.

2. Were there any ductules of human origin seen in the engrafted PHH sites?

This is a fascinating question that we were excited to explore further. To address this comment, we did a CK7/Ku80 co-stain. CK7 is a well-established marker of cholangiocytes/biliary epithelial cells that line the bile ducts. On a very rare occasion, we found Ku80⁺CK7⁺ cells of human origin that appear to contribute to a bile duct structure. This instance is pictured below, highlighted in the inset frame #2. Arrowheads point out CK7⁺Ku80⁺ cells. We also found some CK7⁺ Ku80⁺ cells that appear more like invasive ductular reactions that are common in chronic liver diseases, including alpha-1 antitrypsin deficiency which is the disease this NSG-PiZ mouse strain is modeling. An example of this is pictured below, highlighted in the inset frame #1. Arrowheads point out CK7⁺Ku80⁺ cells. We have not included these results in the manuscript, as we believe it to be beyond the scope of the main message of our paper which is to leverage the mRNA-LNP tool to augment PHH engraftment as a cell therapy to treat liver disease. This is an extremely interesting result nonetheless, and something that we are very intrigued by. This suggests that the cells we transplant are quite plastic, which is a common feature of liver cells during injury and regeneration.

3. Figure 3B shows that the percent of hepatocytes in the successful engraftment are a very large percent of the overall population of mouse and human hepatocytes. The authors should emphasize this in the Discussion.

Thank you for this comment. This point has been emphasized in the Discussion with the following text: “Engrafted PHHs contribute to a large percentage of overall liver area, an impressive and clinically relevant 30%.” Please see page 7 last paragraph.

Reviewer #3 (Remarks to the Author):

Major revision

The article demonstrates the potential of AAV-mediated p21 expression and mitogenic HGF+EGF mRNA-LNP to improve PHH engraftment in two mouse models of human chronic and acute liver diseases, especially in the alpha-1 antitrypsin deficiency (AATD) model. Overall, this is a new strategy for human hepatocyte transplantation. However, there are several issues need be further addressed.

We appreciate the thoughtful feedback from Reviewer #3 and have addressed all comments.

Major points:

1, Previous studies (PMID: 21505264; PMID: 29032169) demonstrated significant hepatocyte repopulation in NSG-PiZ mice, even without proliferative stimulation treatment. It's thus important to determine if the findings in this paper are model-specific. Have similar results been observed in other chronic models?

We are familiar with these two studies mentioned, and referenced them both in our manuscript (Reference #47 and #21). Our references to these studies are as follows:

“The HGF/cMET axis is an appealing target for improving engraftment, as supported by other cell therapy applications using HGF-expressing adenovectors or cMET agonist antibodies for primary mouse and stem cell-derived hepatocytes, respectively^{46,47}.” Please see page 7 third paragraph.

“While notable PHH engraftment has been observed in NSG-PiZ mice, this engraftment was achieved only with drastic preconditioning methods such as partial hepatectomy, monocrotaline injections, or anti-CD95/FAS antibody treatment, which do not reflect clinically relevant approaches²¹.” Please see page 4 first paragraph.

PMID: 21505264: This paper is mouse-to-mouse hepatocyte transplantation, so inherently the kinetics of transplantation and engraftment differ from the work we present here transplanting human cells into mice. Regardless, the authors of this study find the most successful engraftment when using an adenovector to constitutively express HGF as a proliferative stimulus. Meaningful engraftment takes 180 days without ad-HGF and takes 90 days with ad-HGF. This study, therefore, highlights the need for proliferative stimuli to achieve meaningful liver repopulation in a short amount of time, which is needed if this method is to be used as a viable alternative to organ transplantation.

PMID: 29032169: The authors get meaningful engraftment when PHH transplantation is paired with partial hepatectomy, multiple monocrotaline doses, or pre-treatment with an anti-CD95 antibody. These methods all induce liver injury, thus, triggering regeneration/proliferation signals. Intrasplenic engraftment alone is not sufficient to augment normal AAT protein in the sera of mice above the FDA accepted therapeutic threshold, demonstrating the need for improved PHH engraftment via proliferative stimulus.

In summary, both papers compare hepatocyte transplantation with and without proliferative stimuli, and their main message is that we do need proliferative stimuli to achieve faster, greater, and clinically relevant engraftment. In addition, their methods using adenovector HGF, partial hepatectomy, monocrotaline, and anti-DC95 are not likely to be clinically translatable like our method using HGF+EGF mRNA-LNP has the potential to be. These points are covered in the Discussion section of our manuscript. Please see page 7 second and third paragraphs.

2, Regarding the delivery of p21, the authors are encouraged to explain in details the delivery method, duration of action, and its effects. How much p21 is considered effective? And how long these cells should stay p21 positive? Given that many cells are already p21 positive in the disease livers, it is important to determine the optimal dosage and duration of p21 delivery for therapeutic efficacy.

The delivery method of the p21 vector is outlined in the Methods sections as follows: “The AAV8.TBG.PI.p21.WPRE.bGH vector was obtained from the Penn Vector Core. The AAV8.TBG.PI.Null.bGH vector was obtained from Addgene (105536-AAV8). Mice were anesthetized with isoflurane and the vectors were thawed, diluted in sterile PBS, and injected retro-orbitally at a dose of 5.00×10^{11} gc/mouse using 29G Exel International Insulin Syringes (0.5mL).” Please see page 12 second paragraph.

P21 expression induced via AAV8-TBG-p21 has now been quantified in comparison to the control AAV8-TBG-Null as part of Fig. 1a and 1b. The Results section now reads: “We administered one dose of AAV8-TBG-p21 virus that specifically induces p21 expression in hepatocytes, presumably for the lifespan of the mouse as previously reported³⁶. One week after AAV8 injection, nuclear p21 protein was detected in murine hepatocytes (Fig. 1a). We quantified p21⁺ hepatocytes to be on average 15% in the control AAV8-TBG-Null group and 87% in AAV8-TBG-p21 group (Fig. 1b). This recapitulates what is observed in human AATD patients that have a range of p21⁺ hepatocytes up to ~90%, with an average of 12.3% for heterozygous and homozygous patients combined³³. Please see page 4 first paragraph.

We would again like to emphasize that human patients need not be treated with the p21 vector, as they already have lots of p21 expression as discussed in the Introduction section, “Hepatocyte senescence, measured via expression of cell cycle inhibitor p21, is a consistent hallmark of chronic and acute human liver diseases including nonalcoholic steatohepatitis with and without cirrhosis, viral hepatitis with and without cirrhosis, primary sclerosing cholangitis, primary biliary cirrhosis, autoimmune hepatitis, alcoholic steatohepatitis with and without cirrhosis, acute liver failure, acetaminophen overdose, and inborn errors of metabolism like alpha-1 antitrypsin deficiency³²⁻³⁵. Based on the liver disease and its progression, the percentage of p21 positive hepatocytes can reach up to ~90%^{32,33}.” Please see page 3 last paragraph.

For future studies aimed toward clinical translation, it would be important to define more precisely the range of percentage of p21⁺ hepatocytes required in diseased livers for our strategy to be successful in improving cell therapy.

3, The authors should analyze the kinetics of transplantation efficiency in this system, including engraftment and repopulation.

We would like to emphasize that measurement of human albumin (hALB) in serum of mice is widely accepted in the field as the standard method to examine liver repopulation, or liver chimerism, over time. Human albumin secretion data is the ideal surrogate for tracking the transplantation and eventual expansion of human cells over time without the need to sacrifice mice. This same measurement is used for the same purpose by many of the references in this manuscript. We have summarized this point in the Results section as follows: “Serum hALB was measured over time to evaluate the kinetic of PHH transplantation and engraftment globally within the livers, to assess PHH function, and to show the trajectory of human hepatocyte expansion (Fig 2g)” and “Quantification of serum hALB over time is very often used to determine the kinetics of transplantation and engraftment, and as a measure of liver repopulation over time^{12-16,18-21}.” Please see page 5 first paragraph.

4, For the liver delivery of LNP, please provide detailed information regarding the dosage used, as well as the pharmacokinetics (PK) and pharmacodynamics (PD) of the LNP-mediated delivery. Additionally, it is crucial to include data and analysis related to potential hepatotoxicity associated with the LNP treatment.

Dose of mRNA-LNP used is detailed in the Methods sections using the following text: “mRNA-LNP were thawed and freshly diluted on ice in sterile Dulbecco’s Phosphate Buffered Saline (DPBS) prior to each experiment. Mice were anesthetized with isoflurane and administered 50 μ L of diluted mRNA-LNP encoding luciferase (10 μ g/mouse), eGFP (10 μ g/mouse), or combination HGF (5 μ g/mouse) + EGF (5 μ g/mouse) intravenously by retro-orbital injection using 29G Exel International Insulin Syringes (0.5mL).” Please see page 13 second paragraph.

To address the comment regarding PK and PD of LNP-mediated delivery, we consulted with our collaborator and co-author Dr. Ying Tam, CSO at Acuitas Therapeutics. Dr. Tam’s remarks are summarized as follows:

For the PK and biodistribution in both mouse and non-human primates, approximately 60-80% of an intravenous administered mRNA-LNP dose arrives at the liver. On a percent per tissue weight, similar levels reach the spleen and adrenal glands as well. The majority of mRNA-LNP is cleared from the circulation in mouse in <1 hour, while approximately 1-2 hours in the non-human primate. With respect to PD, expression in mice is typically detectable by <1 hour, reaches a maximum by approximately 8 hours, and declines thereafter.

Liver toxicity can be attributed to the dose of lipids and is associated with high mRNA-LNP doses. However, in our studies at the relatively low dose 0.5 mg/kg encapsulated mRNA, we would not expect any detectable liver toxicity. As dose increases around 3-5 mg/kg, effects on the liver are typically shown by dose dependent increases in serum transaminases (ALT, AST), increasing histopathology (hepatocyte vacuolation, single cell and focal necrosis, immune cell infiltration), and some impacts on liver associated clinical chemistry readouts (albumin, alkaline phosphatase) and immunology readouts secondary to liver insult. However, depending on dose, most effects are reversible and return to baseline.

To support the comments above, we conducted further experiments regarding the potential of LNP-induced hepatotoxicity. We injected 10ug of Luc mRNA-LNP into NSG-PiZ mice and analyzed 48 hours later. We measured ALT levels to determine if there is any hepatotoxicity from LNP administration. We also did TUNEL and cleaved caspase 3 stains to examine hepatocyte apoptosis. The results are included in Supplementary Fig. 1a and are summarized in the Results section with the following text: "We confirmed that there was no hepatotoxicity related to mRNA-LNP injections, demonstrated by no change in liver alanine aminotransferase (ALT) enzyme levels (Supplementary Fig. 1a) 48 hours following control firefly luciferase (Luc) mRNA-LNP treatment. We also observed no increase in hepatocyte apoptosis via TUNEL and cleaved caspase 3 staining, which were very minimal prior to mRNA-LNP injection (data not included)." Please see page 4 first paragraph.

5, This liver delivery strategy is clinically significant, and it is important to assess the consistency of LNP delivery efficiency in human hepatocytes, such as in a hepatocyte-humanized animal model.

To address the comment regarding LNP delivery in human hepatocytes, we consulted with our collaborator and co-author Dr. Ying Tam, CSO at Acuitas Therapeutics. Dr. Tam's remarks are summarized as follows:

Efficient delivery of RNA payloads into hepatocytes following IV administration using LNP systems, like those used here in our study, has been extremely well documented in non-human primates in preclinical studies and in humans in clinical trials. For both non-human primates and humans, these include LNP to deliver mRNA and siRNA. To date, the use of chimeric murine models has shown delivery to the engrafted hepatocytes, although the required dose is relatively high in these animals. Work is ongoing in this area.

Acuitas Therapeutics developed the LNP delivery system for Onpattro, a drug to deliver siRNA to the liver to silence a disease-causing gene for mutated transthyretin. Transthyretin is expressed in hepatocytes, and the LNP we designed efficiently delivers the siRNA to these human hepatocytes to silence the gene. This drug was approved in 2018 and its mechanism, including RNA delivery to the liver of humans and non-human primates, is well known and accepted. The paper describing the LNP and development of the drug can be found here (PMID: 31802031). Additionally, this website highlights how Onpattro works: <https://www.onpattrohcp.com/how-onpattro-works>. Given the similarity of the LNP that Acuitas provided us for this study to the ones utilized for Onpattro, we expect our mRNA-LNP to transfect human and mouse hepatocytes with high efficiency.

Similarly, Intellia Therapeutics is a company delivering mRNA encoded gene editing enzymes in very similar LNP delivery systems. One example of their success using LNP delivery system in vitro, in vivo in non-human primates, and in clinical trials in humans can be found here (PMID: 34215024). This web page (<https://www.intelliatx.com/our-science/publications-and-presentations/>) also provides a listing and links to an additional number of presentations and publications on clinical trials showing delivery of mRNA to human hepatocytes via LNP delivery systems.

The above comments and citations highlight the consistency of LNP delivery efficiency in human hepatocytes. It is well established and accepted that LNP delivery is an efficient method to deliver nucleic acid to hepatocytes, highlighting the feasibility of translation of our study.

6, APAP is an acute liver injury model. The authors are encouraged to perform cell transplantation in a clinically related chronic liver disease model.

In this manuscript, we use the NSG-PiZ transgenic mouse model, which recapitulates the chronic liver disease associated with alpha-1 antitrypsin deficiency. The text in our manuscript reads, "As a chronic human liver disease model, we used the NSG-PiZ mouse that recapitulates alpha-1 antitrypsin deficiency (AATD) associated liver disease. These mice are transgenic for the human mutated PiZ allele on immunodeficient NOD scid gamma (NSG) background, allowing them to tolerate xenotransplantation²¹. These mice have heterogeneous accumulation of cytoplasmic misfolded Z-AAT globules in hepatocytes and develop fibrosis with age, similar to human AATD patients^{21,33}." Please see page 4 first paragraph. We performed cell transplantation in this model, and the data from these experiments are presented in Fig. 2 and Fig. 3. Importantly, we show in Fig. 3 that the level of engraftment that we achieve in the best condition is sufficient to reverse the key features of this chronic liver disease such as misfolded protein in the liver, misfolded protein in the blood, and elevated ALT levels. It is well known that this mouse model does not develop significant liver fibrosis until a much later age, thus, we have not explored this chronic liver disease outcome in our study.

Minor points:

1, Absence of control group (Fig. 1a).

The appropriate control AAV8-TBG-Null injected group has been now added to Fig. 1a. In addition, we have quantified the percent of hepatocytes that express p21 in the control and treated groups. This data is now found in Fig. 1b. The text in the Results section reads: "We administered one single dose of AAV8-TBG-p21 virus that specifically induces p21 expression in hepatocytes for the lifespan of the mouse³⁶. One week after AAV injection, nuclear p21 protein was detected in murine hepatocytes (Fig. 1a). We quantified p21⁺ hepatocytes to be on average 15% in the control AAV8-TBG-Null group and 87% in AAV8-TBG-p21 group (Fig. 1b). This recapitulates what is observed in human AATD patients that have a range of p21⁺ hepatocytes up to ~90%, with an average of 12.3% for heterozygous and homozygous patients combined³³." Please see page 4 first paragraph.

2, Lack of NSG age matched and NSG-PiZ age matched groups (Fig. 3b-f).

We believe Figure 3b, 3c, and 3d should not require age matched controls. Age matched controls are non-transplanted mice, thus, a Ku80 stain would be entirely negative (3b), liver repopulation would be 0% (3c), and there would be zero secreted human albumin (3d).

However, as recommended, we have included the NSG age matched and NSG-PiZ age matched groups to Fig. 3e and 3f. The results are summarized as follows: "Not only did we see high levels of PHH engraftment in the experimental group, but also an improvement in the AATD-associated liver disease phenotype. Indeed, repopulation with healthy hepatocytes significantly decreased the 2C1⁺ misfolded Z-AAT polymer area in the liver as compared to the control transplanted group and age-matched NSG-PiZ controls (Fig. 3e, 3f)." Please see page 6 second paragraph.

3, The indicators lack a standard range (Fig. 3g-i).

Standard range for hZ-AAT (3g): Mice do not express human Z-AAT, so there is not a typical range expected. However, as recommended, we used age-matched NSG-PiZ control mice to demonstrate what basal hZ-AAT levels are in this transgenic model. The following text is now included in the Results section: "This decrease was observed when compared to the control transplanted group as well as the age-matched non-transplanted NSG-PiZ mice which express only human Z-AAT via the transgene, hence benchmarking the baseline serum human Z-AAT in this mouse strain (Fig. 3g)." Please see page 6 second paragraph. Furthermore, we have also added

the following text in our Discussion, “There is a growing body of evidence that demonstrates a correlation between circulating Z-AAT polymers and overall liver disease in AATD patients, thus, reducing serum Z-AAT is promising for overall patient outcomes^{49,50}.” Please see page 7 fourth paragraph.

Standard range for hAAT (3h): Mice do not express human AAT, so there is not a typical range expected. What we do know is that the FDA-accepted therapeutic threshold for normal M-AAT protein augmentation is 11uM, which is equivalent to 572 ug/mL. This is the benchmark to achieve when evaluating a therapy for AATD lung disease. It is important to note that the ELISA assay used here detects both misfolded Z-ATT and normal M-AAT. This information has been included in the figure legend for Fig. 3.

Standard range for ALT (3i): As recommended, we have included the following text in the Results section: “As an additional liver function assay, we measured ALT levels, where elevated levels above the normal range for NSG mice (>32 IU/L) are indicative of liver damage³⁹.” Please see page 6 second paragraph.

Reviewers' Comments:

Reviewer #1:

Remarks to the Author:

The response to the reviewers' comments is satisfactory. I recommend its acceptance for publication in the journal Nat. Comm. Meanwhile, I suggest that the authors consider adding a graphical table of contents (TOC) to illustrate the overall research design, facilitating readers' understanding.

Reviewer #2:

Remarks to the Author:

No comments necessary.

Reviewer #3:

Remarks to the Author:

1. While the authors have indeed proposed a proliferation stimulus strategy for cell transplantation, they only validated it in a specialized transplantation model called NSG-PiZ model. In this model, normal hepatocytes naturally have a competitive proliferative advantage, allowing successful repopulation without additional stimuli. To enhance the clinical applicability of this strategy, the authors must evaluate transplantation outcomes in other chronic injury models, even with a short period of repopulation after transplantation.
2. It lacked detailed information on the duration of expression following AAV8-TBG-P21 delivery and the characterization of cells.
3. It is recommended that the authors use hALB staining instead of Ku80 staining to characterize repopulation. In the early stages after hepatocyte transplantation, when the number of engrafted cells is low, relying solely on the secretion of hALB may not provide an accurate assessment.
4. Additional analyses are necessary for LNP-mediated hepatotoxicity. The chosen time points were relatively short, and the analyzed indicators were limited, making it challenging to support the claimed conclusion of no hepatotoxicity.

Author’s point-by-point response (blue) to the reviewers’ comments, reproduced verbatim (black)

We greatly appreciate the comments from the three reviewers to help us improve the quality of the manuscript as well as the editor’s decision to consider our revised manuscript. Here is our point-by-point response to the reviewers’ comments. The main manuscript and supplementary information have been edited accordingly. Changes from the prior submission are highlighted in yellow in the accompanying manuscript file.

Reviewer #1 (Remarks to the Author):

The response to the reviewers’ comments is satisfactory. I recommend its acceptance for publication in the journal Nat. Comm. Meanwhile, I suggest that the authors consider adding a graphical table of contents (TOC) to illustrate the overall research design, facilitating readers’ understanding.

We thank reviewer #1 for considering our revisions as satisfactory and for approving the manuscript for publication in Nature Communications. As recommended, we included the following graphical table of contents/graphical abstract to illustrate the overall research design. This has been included in our manuscript as Supplementary Fig. 1.

Supplementary Fig. 1 | Graphical Abstract. Liver transplantation remains the standard of care for patients with end stage liver disease, however, donor organs are always in limited supply. Alternatively, transplanting primary human hepatocytes (PHH) holds promise for restoring liver function, yet current hepatocyte cell therapy engraftment efficiency is low and long-term benefits are limited. The notion in the field is that to harness effective liver cell therapies, there must be regenerative stimuli and a growth advantage for donor cells. In two mouse liver disease models, one chronic and one acute, we mimic human liver diseases by compromising host hepatocyte proliferation by expressing p21. In these models, we demonstrate that transient, robust expression of human hepatocyte growth factor (HGF) and epidermal growth factor (EGF) in the liver delivered via nucleoside-modified mRNA in lipid nanoparticles (mRNA-LNP) significantly enhances PHH survival, proliferation, and engraftment. Importantly, in the chronic injury model

PHH engraftment is sufficient to reduce disease burden and improves overall liver function. This innovative approach may overcome the current barriers to translating hepatocyte cell therapies, primary or stem cell derived, to the clinic.

Reviewer #2 (Remarks to the Author):

No comments necessary.

We thank reviewer #2 for considering our revisions as satisfactory and for approving the manuscript for publication in Nature Communications.

Reviewer #3 (Remarks to the Author):

1. While the authors have indeed proposed a proliferation stimulus strategy for cell transplantation, they only validated it in a specialized transplantation model called NSG-PiZ model. In this model, normal hepatocytes naturally have a competitive proliferative advantage, allowing successful repopulation without additional stimuli. To enhance the clinical applicability of this strategy, the authors must evaluate transplantation outcomes in other chronic injury models, even with a short period of repopulation after transplantation.

To address this point, we would like to clarify that we have validated our strategy in 2 liver injury mouse models, the chronic NSG-PiZ and the acute APAP injury. Related to the chronic injury model used, we believe that the NSG-PiZ model is in fact not a trivial host environment for human cell engraftment as much as it has been shown for mouse cell engraftment.

Indeed in the mouse[1], Ding et al., a study cited by reviewer #1 and that we discussed in our original manuscript, aim to transplant mouse primary hepatocytes into the PiZ mouse model. In this experimental context, progressive liver repopulation of ~2%, ~10%, and ~70% by transplanted mouse hepatocytes is observed in 30 days, 90 days, and 180 days respectively. In this model, 6 months time is needed to see significant liver repopulation. These data indicate that the diseased PiZ liver environment provides a progressive natural growth advantage to wild type mouse hepatocytes with time. However, it is well recognized that engraftment of mouse cells into a mouse host is much easier to achieve than human cells into a mouse host. This was specifically illustrated in the work from Borel et al., 2017 [2], cited by reviewer #3 and discussed in our original manuscript. This team created the immune deficient NSG-PiZ model allowing human cell transplantation. Engraftment of human primary hepatocytes in the NSG-PiZ mouse was very low without any additional growth stimuli, reaching human serum albumin concentration of ~0.1 mg/mL around 8 weeks. However, when accompanied with drastic additional liver injuries such as partial hepatectomy, treatment with anti CD95 antibody, or 2 injections of the hepatotoxic monocrotaline, repopulation was greatly improved and human serum albumin levels reached ~1 mg/ml around 8 weeks, which then plateaued for an additional few weeks. The authors report a maximum of 25% liver repopulation with human cells at 10 weeks in their best condition pretreated with monocrotaline twice. In our manuscript, we confirmed similar low engraftment of human primary hepatocytes in NSG-PiZ mice after 5 weeks in the absence of growth stimuli or additional injury. Human ALB+Ku80+ cells occupied ~2% of liver surface area, and secretion of human serum ALB remained low at 0.1 mg/mL. Here we showed that HGF+EGF mRNA-LNP treatment for 5 weeks improves engraftment, where up to ~30% of liver surface area is occupied by human ALB+Ku80+ cells which is a 14.13-fold increase compared to control mice. There is a corresponding 37.15-fold increase in human serum ALB compared to control mice, reaching values of ~5mg/mL, which were significantly higher than those reported by Borel et al. Overall, our study using this chronic liver injury NSG-PiZ model not only confirms the low engraftment of

human primary hepatocytes in this model, as reported previously [2], but importantly proposes a clinically relevant protocol to rapidly and significantly promote engraftment of human cells and mitigates the liver disease burden.

For these reasons, we believe that the NSG-PiZ model is not such a specialized and trivial liver environment for human cell engraftment, and rather recapitulates some key clinical features of many human chronic liver diseases including compromised hepatocyte proliferation, low degree of hepatocyte apoptosis and low turnover of hepatocyte proliferation. Given these features, the NSG-PiZ model offers a slight growth advantage to human donor healthy cells, but this alone is not sufficient to see significant engraftment of human hepatocytes. Most of the liver injury models used in the field for cell therapy assessment employ drastic liver injuries where vast hepatocyte ablation is induced through partial hepatectomy, hepatotoxic treatments such as monocrotaline or anti CD95 antibody, the uPA transgenic model, and or using lethal genetic models such as FAH^{-/-} mice and HSVtk transgenic mice [3]. These models are rather artificial, induce ablation of a majority of hepatocytes, and do not recapitulate human liver disease as the NSG-PiZ model does. We thus believe that a third liver injury model, using one of those listed models, would only incrementally advance the clinical relevance of our protocol for improving cell engraftment.

2. It lacked detailed information on the duration of expression following AAV8-TBG-P21 delivery and the characterization of cells.

This is an important point that we addressed by including an additional time point for p21 expression in hepatocytes 6 weeks after AAV-p21 injection.

In Fig.1, we show that AAV8-TBG-p21 injection induces p21 expression in a majority of host hepatocytes after 1 week. In addition, we have now performed an immunostaining for p21 five weeks after cell transplantation or 6 weeks after administration of the AAV8 vectors. This data has been included in the manuscript as Supplemental Fig. 4. The following text has also been included in the manuscript results “At this time, p21 expression was observed in 6% of hepatocytes in the control group and 21% in the treated group (Supplementary Fig. 4a, b). This relative decrease in the percentage of p21+ hepatocytes as compared to Figure 1b can be attributed to the fact that a large portion of the livers in the treated group are now occupied by p21- donor PHHs which are on average smaller than neighboring mouse hepatocytes. The remaining p21+ cells are mouse host hepatocytes as expected.” The persistent expression of p21 in mouse hepatocytes after AAV8 delivery is in line with previous reports showing protein expression for at least 9 months after AAV8 injection in mice [4].

The p21 expressing hepatocytes are mouse hepatocytes that contain, for most of them Z-AAT globules, as indicated with the visible globular red fluorescent background in Fig. 1a.

3. It is recommended that the authors use hALB staining instead of Ku80 staining to characterize repopulation. In the early stages after hepatocyte transplantation, when the number of engrafted cells is low, relying solely on the secretion of hALB may not provide an accurate assessment.

We apologize if the main text and figure legends were not explicit enough as we have indeed consistently evaluated cell engraftment by measuring the secretion of human serum ALB over time at 2 weeks (Fig. 2g) and 5 weeks (Fig. 3d) after cell transplantation, and by quantifying the engrafted cell surface area co-stained for hALB and Ku80 on liver sections at these time points, as the reviewer suggested. The liver tissue surface area occupied by engrafted human cells was measured via quantification of hALB+ Ku80+ double positive cells, which is illustrated in graphs

for both time points. As indicated in our revised manuscript, secretion of human serum albumin and measuring surface area of engraftment, are widely used and accepted in this field. These data are included in the following figure panels:

- Fig. 2b for hALB/Ku80 co-staining images @ 2 weeks
- Fig. 2f for quantification of hALB+ Ku80+ area @ 2 weeks
- Fig. 3b for Ku80 staining images @ 5 weeks
- Supplementary Fig. 3b for hALB/Ku80 co-staining images @ 5 weeks (also shows split channel images to clearly highlight that transplanted cells consistently stain positive for hALB and Ku80)
- Fig. 3c for quantification of hALB+ Ku80+ area @ 5 weeks

To better clarify our systematic two ways of measuring cell engraftment - by quantifying hALB+ Ku80+ areas and human serum ALB secretion levels - , we changed the title of the y axis of both graphs (Fig. 2f and Fig. 3c). The new axes now read “% hALB+ Ku80+ areas” instead of “% PHH repopulation”. We have also included fold increase of PHH repopulation compared to control mice in graphs as we did for the serum hALB data for both time points. Importantly, these fold-increases obtained by both assays are similar and thus confirm the robustness of our protocol to promote engraftment.

4. Additional analyses are necessary for LNP-mediated hepatotoxicity. The chosen time points were relatively short, and the analyzed indicators were limited, making it challenging to support the claimed conclusion of no hepatotoxicity.

We agree with the reviewer that we only discussed hepatotoxicity of LNP 48 hours after LNP injection in supplementary Fig. 2a. That was assessed by measuring serum ALT levels, which is a well-accepted assay to evaluate liver damage. To specifically address this point, we also now included a discussion of the data from Fig. 3i in the manuscript text. This figure panel shows ALT levels at the 5 weeks-time point. These data indicate that ALT levels in the control NSG-PiZ mice, representing mice treated with control mRNA-LNP (Luc mRNA-LNP) and control AAV8 (AAV8-TBG-NUL) had statistically similar levels of ALT compared to non-injected NSG-PiZ mice, indicating no additional liver damage from 10 mRNA-LNP injections. In addition, we have included a second toxicity assay. Cell apoptosis has been assessed with TUNEL assays at the two time points, 48 hours time point after 1 injection of control LNP and 5 weeks time point after 10 injections of control LNP. Cell apoptosis remains very low to undetectable in all groups, controls and treated. These data have been included in Supplementary Fig. 2 and Supplementary Fig. 7.

References:

1. Ding, J., et al., *Spontaneous hepatic repopulation in transgenic mice expressing mutant human α 1-antitrypsin by wild-type donor hepatocytes*. J Clin Invest, 2011. **121**(5): p. 1930-4.
2. Borel, F., et al., *Survival Advantage of Both Human Hepatocyte Xenografts and Genome-Edited Hepatocytes for Treatment of alpha-1 Antitrypsin Deficiency*. Mol Ther, 2017. **25**(11): p. 2477-2489.
3. Du, Y., et al., *Mouse Models of Liver Parenchyma Injuries and Regeneration*. Front Cell Dev Biol, 2022. **10**: p. 903740.
4. Zincarelli, C., et al., *Analysis of AAV serotypes 1-9 mediated gene expression and tropism in mice after systemic injection*. Mol Ther, 2008. **16**(6): p. 1073-80.